

# Co-evolution of terrestrial and aquatic ecosystem structure with hydrological change in the Holocene Baltic Sea

Gabriella M. Weiss*[1,a,b], Julie Lattaud*[2], Marcel T. J. van der Meer[1], Timothy I. Eglinton[2]

[1]Department of Marine Microbiology and Biogeochemistry, The Royal Netherlands Institute for Sea Research (NIOZ), Texel, The Netherlands
[2]Swiss Federal Institute of Technology of Zurich (ETHZ), Biogeoscience Group, Zurich, Switzerland

Present address:
[a]Astrobiology Center for Isotopologue Research, Department of Geosciences, Pennsylvania State University, State College, PA, USA
[b]Division of Geological and Planetary Sciences, California Institute of Technology, Pasadena, CA, USA

*Both authors contributed equally to this work

*Correspondence to*: Gabriella M. Weiss (gweiss@caltech.edu) , Julie Lattaud (jlattaud@ethz.ch)

**Abstract.** The Baltic Sea experienced a number of marine transgressions and regressions throughout the Holocene. These fluctuations in sea level coupled with substantial regional ice melt led to isostatic adjustment and periodic isolation from the North Sea. Here, we determine the distributions and isotopic signatures of organic compounds preserved in a sediment record spanning the last ~ 11 ka in order to reconstruct environmental change under these dynamic conditions. Carbon and hydrogen isotope ratios of short-, mid-, and long-chain *n*-alkanes along with long-chain diol and glycerol dialkyl glycerol tetraether abundances were analyzed from Arkona Basin sediments sampled from the western Baltic Sea. In the earliest part of the record (10 – 8.2 ka), hydrogen isotope values of higher plant-derived *n*-alkanes revealed a change in dominant water source from an ice melt-derived to a precipitation-dominated hydrological regime. Following this shift in water source, carbon isotope values of *n*-alkanes suggest diversification of vegetation. Shifts in hydrology and vegetation did not coincide with established phase boundaries, but instead occurred mid-phase or spanned phase transitions, highlighting the fact that proxies may record changes on different time scales and suggesting that climate in the region was dynamic throughout the Holocene.

## 1 Introduction

The Baltic Sea, located in Northwest Europe, is a semi-enclosed marine basin characterized by restricted circulation, strong terrestrial influence, and relatively low oxygen concentrations that promote enhanced sedimentary organic matter contents and diverse phytoplankton communities. Today there is an open connection between the North Sea and the Baltic Sea via the Danish Straits in the west; this connection has not been permanent, causing the basin to experience substantial environmental change throughout the Holocene. The Baltic Sea was formed during the last deglaciation as a consequence of the melting of the Scandinavian Ice Sheet (SIS) that covered large swaths of Europe. The Baltic Ice Lake – the first manifestation of the



Baltic Sea – emerged prior to 13 ka as large portions of the SIS melted, exposed the land, and formed a meltwater lake filled with icebergs (Björck, 1995). As global and regional temperatures continued to rise during the Holocene, rising sea levels led

to an inundation of saline North Sea water into the lake, resulting in the transition of the basin from a freshwater to a brackish lake, a phase known as the Yoldia Sea (YS) (11.2 to 10.6 ka). The retreat of the SIS and associated isostatic rebound of surrounding landmass resulted in shoreline and sea level fluctuations (e.g., Moros et al., 2002), which led to another low salinity phase known as the Ancylus Lake (AL). The AL phase terminated at around 7.7 ka when a large marine transgression, lasting for around 500 years (7.7 to 7.2 ka), reconnected the Baltic Sea with the North Sea. This resulted in a second brackish

phase called the Littorina Sea (LS), which lasted until 3 ka. After 3 ka, the Modern Baltic phase was established (Andrén et al., 2000, 2011), with further freshening due to continued continental uplift and lack of large marine transgressions. The complex climate dynamics caused substantial shifts in the salinity of the Baltic Sea during the Holocene, indicated by changes in the phytoplankton population (Alhonen, 1972; Weiss et al., 2020).

Biomarkers, organic compounds indicative of a particular environmental process or produced exclusively by a particular group

of organisms, are widely used organic geochemical tools that provide a wealth of paleoclimatic information. An overview of common proxies and the parameters they indicate can be found in Table 1. Historically, geochemists have made use of isotope measurements of bulk sediments or biomass, which present a general overview of a given system (e.g., Leng and Lewis, 2017). Compound−specific carbon isotope ratios ($\delta^{13}C$, the ratio of $^{13}C/^{12}C$ in a sample relative to a standard, Vienna Pee Dee Belemnite, aka VPDB) of organic molecules give insight into sources of organic matter (e.g., terrestrial versus marine; Meyers,

1994) and atmospheric $CO_2$ conditions (e.g., Freeman and Hayes, 1992). Compound−specific hydrogen isotope ratios ($\delta^2H$, the ratio of $^2H/^1H$ in a sample relative to a standard, Vienna Standard Mean Ocean Water, aka VSMOW) of lipids permit further understanding of compound biosynthesis and environmental conditions, such as precipitation and salinity (e.g., Sachse, Billault et al., 2012). These two elemental signatures are widely applied to study past environmental change across the planet and throughout the geologic record (e.g., Tipple and Pagani, 2010; Schefuß et al., 2011; Sinninghe Damsté et al., 2011).

Straight-chain hydrocarbons, such as *n*-alkanes, are especially resistant to degradation and alteration over geological timescales and are found as far back as the Precambrian (McKirdy and Hahn, 1982; Gordadze et al., 2017). In addition to being relatively recalcitrant, *n*-alkanes are ideal biomarkers for a C and H isotope approach because they contain non-exchangeable C and H, and are synthesized by a variety of organisms from bacteria to higher plants (e.g., Hemingway et al., 2016). Generally, long-chain *n*-alkanes are found in the protective wax layer of leaves from terrestrial higher plants (Eglinton and Hamilton, 1967),

and the C isotopic compositions of *n*-alkanes from these plants can differ by up to 20‰ depending on metabolism ($C_3$, $C_4$, Crassulacean Acid Metabolism, aka CAM; see Diefendorf & Freimuth, 2017 for review). $C_4$ plants produce more $^{13}C$-enriched *n*-alkanes than $C_3$ plants, and *n*-alkanes synthesized by CAM plants show a wide range of isotopic compositions (Chikaraishi and Naraoka, 2003, 2005, 2007; Diefendorf and Freimuth, 2017). Mid-chain *n*-alkanes ($C_{23}$ and $C_{25}$) are mainly produced by submerged macrophytes in freshwater and marine environments (Ficken et al., 2000), and have characteristic C and H

signatures indicative of their aquatic origins (Aichner et al., 2010, 2017). H isotopes of *n*-alkanes longer than $C_{21}$ reflect plant growth water at the time of leaf formation, making them useful for paleohydrology studies (e.g., Sternberg, 1988; Sachse et





al., 2012; McFarlin et al., 2019). Shorter-chain $n$-alkanes ($< C_{21}$) are principally produced by phytoplankton and bacteria (Meyers and Ishiwatari, 1993; Zhang and Sachs, 2007), and their H isotopic composition also correlates strongly with growth water (Sachse et al., 2004). Moreover, $n$-alkanes can be transported to sediments as aerosolized particles that are deposited

directly in coastal waters and lakes or transported by rivers (Schefuss et al., 2005). The spatial coverage of this atmospheric transfer is dependent upon climate conditions and density of vegetation, but $n$-alkanes are generally deposited within weeks after being aerosolized (e.g., Nelson et al., 2018), making them ideal for paleoenvironmental reconstructions.

In addition to $n$-alkanes, other lipid biomarkers – such as glycerol dialkyl glycerol tetraethers (GDGTs) and long-chain diols (LCDs) – preserve information about the environmental conditions under which they were produced. An abundance of

branched GDGTs (brGDGTs) reflects bacterial communities – believed to be Acidobacteria – in soils (Weijers et al., 2007, 2009; Sinninghe Damsté et al., 2018), thus should be representative of terrestrial environments or terrestrial input at aquatic sites. However, aquatic in situ production convolutes the strictly terrestrial interpretation of brGDGTs (De Jonge et al., 2014, 2015). Luckily, the process is characterized by a number of indices that distinguish between allochthonous and autochthonous production (Hopmans et al., 2004; Sinninghe Damsté, 2016; Xiao et al., 2016; Martin et al., 2020). LCDs are synthesized by

both marine and freshwater phototrophic organisms (Volkman et al., 1992), and have been used to trace freshwater input into coastal environments using one isomer, the $C_{32}$ 1,15-diol (Lattaud et al., 2017). LCDs are also used to reconstruct sea surface temperature via the long-chain diol index (LDI, Rampen et al., 2012; de Bar et al., 2020). In addition, the presence of 1,14- and 1,12-diols are characteristic of diatom abundance, more precisely of the presence of *Proboscia* spp. (de Bar et al., 2020; Rampen et al., 2007, 2014). The presence of *Proboscia* diatoms has been shown to hinder LDI temperature reconstructions

due to their contribution to the pool of 1,13-diols used in the index. At a minimum, pairing brGDGT with LCD data can contribute to understanding of complex riverine transport of organic matter and soil erosion in a given location (Lattaud et al., 2017), and in some environments can help reconstruct sea surface temperature.

Recently, attempts to understand the complex salinity history of the Baltic using H isotope values of long-chain alkenones highlighted hydrological shifts that did not coincide with a change in the phytoplankton population in the Early Holocene

(Weiss et al., 2020). Long-chain alkenones are a species-specific biomarker produced by haptophyte algae that have been widely used to discern changes in sea surface temperature and salinity (e.g., Brassell et al., 1986; Weiss et al., 2019). This observation, in conjunction with previous studies of Baltic Sea climate history, serves as the basis for the present investigation, which aims to compare information gathered from terrestrial (higher plant) biomarkers and aquatic biomarkers with the data from GDGTs and long-chain diols in order to delineate regional effects of large-scale hydrological changes.

**2 Methods**

Expedition 64PE410 aboard the RV *Pelagia* retrieved a 12 m-long piston core from the Arkona Basin (64PE410_S7, 54°55.208 N, 13°29.992 E, Fig. 1) in 2016 to study the climate history of the marginal marine environment of the Baltic Sea. The Arkona basin is situated in the western part of the Baltic Sea, where saline North Sea water enters into the basin (Fig. 1). The age

model is described in detail by Weiss et al. (2020) and was created by combining $^{14}$C-ages of mollusk shells and correlation

of Ca/Ti and Br records with a nearby core (Warden et al., 2016).

**2.1 Bulk isotope analyses**

Samples of bulk sediment were freeze-dried for analyses every 100 cm at the top of the core and every 10 cm after 930 cm. An aliquot of 10 to 15 mg of freeze-dried sediment was sampled for total organic carbon (TOC), total nitrogen (TN), and bulk carbon ($\delta^{13}$C) and nitrogen ($\delta^{15}$N) isotope composition. Aliquots were acidified with 2 M hydrochloric acid and rinsed with

deionized water to remove carbonates and salts. Isotopic composition and elemental weight percentage of bulk organic matter were measured with an elemental analyzer (EA, Thermo Scientific Flash 2000) coupled to an isotope ratio mass spectrometer (IRMS, Thermo Scientific Delta V).

**2.2 Lipids**

Lipids were extracted as described in Weiss et al. (2020). Briefly, freeze-dried sediment samples (11 – 17 grams) were

extracted via an accelerated solvent extractor (ASE 350, Dionex, Thermo-Scientific, Sunnyvale, CA, USA) with dichloromethane (DCM) : methanol (MeOH) 9 : 1 (*v* / *v*). Internal standards – C$_{46}$ GDGT (Huguet et al., 2006) C$_{22}$ 7,16-diol, C$_{19}$ ketone, and C$_{36}$ *n*-alkane – were added to all total organic extracts. The extracts were dried under a stream of N$_2$ and separated over an aluminum oxide column (activated for 3 h at 150 °C). An apolar fraction (containing *n*-alkanes) was eluted with four column volumes of hexane : DCM 9 : 1 (*v* / *v*), a ketone fraction with three column volumes of hexane : DCM 1 : 1

(*v* / *v*), and the polar fraction (containing glycerol dialkyl glycerol tetraethers, GDGTs, and long-chain diols, LCDs) with three column volumes of DCM : MeOH 1 : 1 (*v* / *v*).

**2.2.1 *n*-Alkanes**

The *n*-alkanes were quantified based on a known concentration of a C$_{36}$ *n*-alkane standard on a HP 7890A gas chromatograph (GC) equipped with a flame ionization detector (FID) and a VF-1 ms capillary column (Agilent, 30 m × 0.25 mm × 0.25 μm).

The temperature program started with a 1 min hold at 50 °C, followed by a 10 °C min$^{-1}$ ramp to 320 °C, and held at 320 °C for 5 min. Reported concentrations are normalized to TOC content (ng g$_{TOC}^{-1}$).

**2.2.2 Long-chain diols (LCDs)**

Aliquots of the polar fraction were silylated with N,O-Bis(trimethylsilyl)trifluoroacetamide (BSTFA) and pyridine (10 μL each) at 60 °C for 30 min, after which ethyl acetate was added. Long-chain diols (LCDs) were analyzed following Rampen et

al. (2012) by GC (Agilent 7990B GC) coupled to a mass spectrometer (Agilent 5977A MSD; GC-MS) equipped with a fused silica capillary column (Agilent CP Sil-5, 25 m × 0.32 mm × 0.12 μm). The temperature program was as follows: started at 70 °C, increased to 130 °C at 20 °C min$^{-1}$, increased to 320 °C at 4 °C min$^{-1}$, held at 320 °C for 25 min. Flow was held constant



at 2 mL min$^{-1}$. The LCDs were identified via single ion monitoring (SIM) of the $m/z$ = 299.3 (C$_{28}$ 1,14-diol), 313.3 (C$_{28}$ 1,13-diol, C$_{30}$ 1,15-diol), 327.3 (C$_{28}$ 1,12-diol, C$_{30}$ 1,14-diol), and 341.3 (C$_{30}$ 1,13-diol, C$_{32}$ 1,15-diol) ions ( Versteegh et al., 1997; Rampen et al., 2012; de Bar et al., 2020). LCDs were quantified based on a known concentration of the C$_{22}$ 7,16-diol.

### 2.2.4 Glycerol dialkyl glycerol tetraethers (GDGTs)

For GDGT analyses, an second aliquot of the polar fraction – at a concentration of 2 mg mL$^{-1}$ – was dissolved in hexane : isopropanol (IPA) 99:1 ($v / v$) and analyzed using ultra high performance liquid chromatography mass spectrometry on an Agilent 1260 UHPLC coupled to a 6130 Agilent MSD following Hopmans et al. (2016). The Branched and Isoprenoid Tetraether (BIT) index (Hopmans et al., 2004) was calculated as follows:

$$BIT = \frac{GDGT-I + GDGT-II + GDGT-III}{GDGT-I + GDGT-II + GDGT-III + crenarchaeol} \tag{1}$$

where roman numerals refer to the number of methyl groups attached to the 5 or 5' positions of the alkyl chain (zero, one, and two, respectively). GDGT concentrations were calculated based on a known concentration of the C$_{46}$ GDGT standard (Huguet et al., 2006).

The #ring$_{tetra,}$ used to assess the source of brGDGT in the marine water column, where values higher than 0.7 indicate sedimentary in situ production (Sinninghe Damsté, 2016) was also used. #ring$_{tetra}$ is defined as follows:

$$\#ring_{tetra} = \frac{GDGT-Ib + 2 \times GDGT-Ic}{GDGT-Ia + GDGT-Ib + GDGT-Ic} \tag{2}$$

The roman numeral I refers to the number of methyl groups, as indicated for the BIT index. The letters 'a', 'b', and 'c' refer to the number of cyclopentane moieties (zero, one, and two, respectively).

### 2.3 Compound-specific isotope analyses

Compound–specific carbon isotope analyses of $n$-alkanes ($\delta^{13}C_{alkane}$) were performed in duplicate on a Thermo Trace GC (1310) coupled with a Thermo Delta-V plus IRMS at the Climate Geology Department of the ETH Zurich. The GC was equipped with a RTX-200 60 m capillary column (Restek, 60 m × 0.25 mm × 0.25 µm). The temperature program was as follows: ramp from 40 °C to 120 °C at 40 °C min$^{-1}$, followed by a 6 °C min$^{-1}$ ramp to 320 °C, held at 320 °C for 12 min.

Hydrogen isotope ratios of $n$-alkanes ($\delta^{2}H_{alkane}$) were measured at the Royal Netherland Institute for Sea Research in duplicate by GC coupled to an IRMS via a high temperature conversion reactor. The GC was equipped with a CP Sil-5 column (Agilent, 30 m × 0.25 mm × 0.25 µm). The temperature program was as follows: started at 70 °C, increased to 130 °C at 20 °C min$^{-1}$, increased to 320 °C at 4 °C min$^{-1}$, held at 320 °C for 10 min. At the start of each day, the H$_3^+$ factor was measured and corrected for (Sessions et al., 2001). H$_3^+$ values ranged from 4.3 to 4.6. Prior to measurement of samples for both C and H isotopes, a $n$-alkane mixture (Mix B, supplied by A. Schimmelmann, Indiana University) was measured. Sample analyses were only



conducted when the average difference and standard deviation between online and offline values was less than 5‰ for H and 0.5‰ for C. For both C and H isotope measurements, the reported standard error is based on duplicate analyses.

## 3 Results

The core covers the last 10.6 ka with average sedimentation rates between 55 and 250 cm ky$^{-1}$ (Weiss et al., 2020). The large variations in sedimentation rates for core 64PE410_S7 and the nearby cores to which it has been correlated are likely related to the shallow water depth in the Arkona Basin (~ 45 m) as well as changes in connection with the North Sea. Bulk isotopes ($\delta^{13}C_{bulk}$ and $\delta^{15}N_{bulk}$ values, as well as TOC and TN content), lipid concentrations, and compound-specific $n$-alkane isotope analyses were conducted on 22 samples.

### 3.1 Bulk isotopes

TOC and TN ranged from 0.5 to 7.2 and 0.1 to 0.8, respectively, with higher values in the most recent part of the core (Fig. S1). $\delta^{13}C_{bulk}$ values varied by ~3‰ in the Early Holocene (< 7.7 ka) followed by a small shift after the marine transgression (Fig. S1). $\delta^{15}N_{bulk}$ values were elevated at the transition from the YS to the AL (5.5‰) followed by a drop to 2.8‰ in the early AL phase (10.1 ka). Values reached 4‰ during the latter half of the AL, succeeded by a second drop at the marine transgression
(2.8‰). Values slowly increased during the early LS phase. Values of 4 ± 1‰ occurred across the MB phase (Fig. S1). Elemental C and N (C/N) ratios ranged from 5.6 to 13.4, spanning common values for both marine and terrestrial ecosystems (Thornton and McManus., 1994). C/N ratios are plotted against $\delta^{13}C_{bulk}$ values for all samples; a mix of freshwater, marine, and terrestrial C3 plant-derived values are recorded, which group by established Baltic Sea phases (Fig. 2).

### 3.2 $n$-alkane distributions and isotopic composition

Organic carbon-normalized concentrations of $n$-alkanes ($C_{21}$ to $C_{35}$) were highest in the older part of the record (2.8 µg g$_{TOC}$$^{-1}$ at 10.6 ka, Fig. 3), followed by a decrease until 9.4 ka (with a notably low concentration at 10.2 ka). The relative abundance of $n$-alkane homologues is comparatively invariant across the record (Fig. 3), except for a slight increase in the abundance of the $C_{21}$ $n$-alkane following the marine transgression and absence or low abundance of the $C_{17}$ $n$-alkane during the AL. The most abundant $n$-alkane is the $C_{29}$ (21 ± 4 %, n=23), followed by the $C_{27}$ (19 ± 2 %, n=23), $C_{25}$ (16 ± 3 %, n=23) and $C_{31}$ (16
± 3 %, n=23). C and H isotope values are plotted as weighted averages for short-chain ($C_{21}$), mid-chain ($C_{23}$ to $C_{25}$), and long-chain ($C_{27}$ to $C_{31}$). Both $\delta^{13}C_{alkane}$ and $\delta^2H_{alkane}$ values captured substantial fluctuations during the early part of the record (before ca. 7.7 ka), but show relatively stable signals for both following the marine transgression, likely the result of low sampling resolution (Tables 2 and 3; Figs. 4 and 5). Weighted-average $\delta^{13}C$ values of the mid- and long-chain $n$-alkanes were similar from 10.6 to 8 ka, after which the mid-chain $n$-alkanes shifted toward higher values and the long-chain $n$-alkanes remained
similar (Fig. 4). The LEaf-Wax Isotopic Spread (LEWIS) index reflects changes in plant species diversity (Magill et al., 2019) and is calculated as follows:



$$LEWIS = \max \delta^{13}C_{23-31} - \min \delta^{13}C_{23-31} \hspace{4cm} (3)$$

Values varied from 2.0 ± 0.7 ‰ between 11 – 7.7 ka to 6.7 ± 1.4 ‰ between 7.3 – 1.3 ka (Fig. 6a). H isotope values of short-, mid-, and long-chain $n$-alkanes tracked the same trends across the record. The $\delta^2H$ values of all $n$-alkanes were higher in the

oldest part of the record (YS, prior to 10.6 ka), and recorded a rapid decrease of ~50‰ at the onset of the AL phase (Fig. 5). Subsequently, a second, considerable increase of ~50‰ occurred between 10 ka and 9.2 ka, succeeded by a gradual decrease lasting until the marine transgression at 7.7 ka. $\delta^2H$ values showed higher values across the marine transgression. Data is limited for the LS phase, but no large isotopic shifts were noted in the samples analyzed here.

### 3.3 Abundance of LCDs and BIT index

Low concentrations of LCDs were detected in a majority of samples prior to the marine transgression. Furthermore, no 1,12-diols were detected in the core indicating limited input from *Proboscia* diatoms (de Bar et al., 2020). The $C_{28}$ 1,14-diol was only present during the MB phase. For all other LCDs, the main change in abundance (relative to all LCDs) occurred at 7.9 ka with a drop in $F_{C32\ 1,15-diol}$ (F = fractional abundance, from 20 to 1%) and an increase in $F_{C30\ 1,15-diol}$ (30 to 80%). LDI-temperature reconstructions (Table S1) were similar to those of Kotthoff et al. (2017) with relatively high temperatures during the YS (18.5

± 0.2 °C) and a decrease during the AL (11.5 ± 2.4 °C). Maximum temperatures occurred during the LS phase, in agreement with the Holocene thermal maximum in Europe (21.5 ± 1.5 °C, 24 °C at 7.2 ka). Temperatures during the MB were similar to modern summer temperatures (17.2 ± 0.1 °C). The coldest period (AL) is also the period with the highest proportion of $C_{32}$ 1,15-diol.

The distribution of brGDGTs changed drastically across phases (Fig. S2) with a distribution similar to soils for all phases

except the AL, which suggests mixing between lacustrine and soil or permafrost-derived brGDGTs. BIT values were higher in the older part of the record (0.8 before 9.9 ka), then continuously decreased to 0.1 at 6.5 ka. Similar BIT values were calculated for all LS samples (Fig. 6). #ring$_{tetra}$ was lower than 0.7 before the transgression (0.5 ± 0.1). A rapid increase occurred at the transgression (~7.7 ka) and values remained consistently high afterwards (0.7 ± 0.1, Table S2).

### 4 Discussion

Compilation of biomarker distributions with corresponding $n$-alkane C and H isotope compositions provided new insights into changes in the regional hydrology and vegetation in the western Baltic region throughout the Holocene. Isotope and biomarker data are discussed below by Baltic Sea phase.

### 4.1 Yoldia Sea

The record covers only a brief snapshot of the end of the YS phase, a period in Baltic Sea history thought to have higher

salinity as evidenced by the presence of marine phytoplankton species (Sohlenius et al., 1996). The presence of short-chain $C_{17} – C_{21}$ $n$-alkanes is suggestive of aquatic in situ production, and the $C_{17}$ $n$-alkane in particular is produced by algae (Gelpi



et al., 1970; Meyers and Ishiwatari, 1993). The only LCDs present are the $C_{30}$ 1,13-, $C_{30}$ 1,15-, and the $C_{32}$ 1,15-diols. LCDs are produced by marine and freshwater eustigmatophyte species (Balzano et al., 2018; Rampen et al., 2014; Volkman et al., 1992). The presence of the $C_{32}$ 1,15-diol, produced by eustigmatophyte species in rivers and stagnant freshwater pools (e.g.,

Lattaud et al., 2017; Balzano et al., 2018), implies low salinity to freshwater conditions. Corresponding $\delta^{13}C_{bulk}$ values versus C/N ratios for this phase fall just on the edge of values common for freshwater ecosystems (Fig. 2), and along with the presence of the $C_{32}$ 1,15-diol, suggest that salinity of the basin may have already significantly declined by the end of the YS phase. In fact, some records divide the YS into two parts: an early marine phase followed by a freshwater phase (e.g., Sohlenius et al., 1996). The highest $n$-alkane concentrations of the whole record, together with highest $\delta^2H_{alkane}$ values, occur just at the phase

boundary (Fig. 3 and 5). The increase in $\delta^2H_{alkane}$ values varies from around 12‰ ($C_{21}$ and $C_{23}$ $n$-alkanes) to 47‰ ($C_{29}$ $n$-alkane), and may be linked to regional climate conditions – likely increased warming and meltwater – which drove the shift from the YS phase into the low salinity AL phase.

### 4.2 Ancylus Lake

The AL phase was the most dynamic of the whole record and includes a number of large isotopic shifts. C isotope compositions

of mid- and long-chain $n$-alkanes follow similar trends during the AL. A series of C isotope fluctuations of around 2‰ were recorded during the AL. There are two shifts to more negative values: one at 10.2 ka and one at 8.2 ka, and an increase at 9.2 ka. The $n$-alkanes and their C isotope signatures imply largely terrestrial, higher plant input during the AL, supported by the absence of the aquatically produced $C_{17}$ $n$-alkane during most of this phase (Fig. 3). The LEWIS index values are the lowest of the record (Fig. 6a), suggesting a lack of species diversity (Magill et al., 2019). This is supported by nearby pollen records

that indicate the same absence of higher plant diversification, likely due to the relatively cold temperatures in Northern Europe at this time (Seppä and Birks, 2001; Seppä et al., 2005; Antonsson et al., 2006), which explains the similarity of $n$-alkane C isotope values. In contrast, the $\delta^{13}C$ values of $C_{21}$ $n$-alkane were higher compared with the longer-chain (> $C_{27}$) $n$-alkanes (-29 versus -21‰ before 7.8 ka), suggestive of mixed terrestrial and aquatic production or emergent aquatic productivity (Ficken et al., 2000).

#### 4.2.1 Ancylus Regression

At 10.2 ka, the Baltic Sea experienced a large regression (Moros et al., 2002), a likely reflection of continental uplift that resulted in shallowing and freshening of the basin. C isotope values of all $n$-alkanes record a decrease at the time of the AL regression and LEWIS index values are low, suggesting low diversity. Low concentrations of $n$-alkanes (~150 ng $g_{TOC}^{-1}$, Fig. 3) occurred at the AL regression, implying less continental runoff into the Arkona Basin at this time. Simultaneously, there

was a 15 – 20‰ increase in $\delta^2H_{alkane}$ values (Fig. 5). Following the Ancylus Regression, $\delta^2H_{alkane}$ values become lower, coincident with a rapid, order of magnitude increase in concentration of $n$-alkanes (Fig. 3 and 5). The SIS was retreating at this time (Muschitiello et al., 2015; Cuzzone et al., 2016), thus it is plausible that a meltwater pulse transported a higher concentration of $n$-alkanes from the north into the basin just after 10.2 ka. A coincident increase in the BIT index from 0.5 to



0.8 (Fig. 6b) and #ring$_{tetra}$ values below 0.7 (suggesting a lack of in situ production, Sinninghe Damsté, 2016), strengthen the
argument for increased input of allochthonous material into the basin. The higher proportion of GDGT-III (Table S2) is indicative of combined in situ brGDGT production and permafrost or soil erosion (Russell et al., 2018; Warden et al., 2018; Kusch et al., 2019). No major changes appear to have occurred in the water column, but progressive melting of the SIS likely caused transport of soil brGDGTs leading to the increase in BIT index values. Furthermore, a peak in F$_{C32\ 1,15\text{-diol}}$ to around 50% likely indicate an increased freshwater runoff (or a decrease in other diols) as the $C_{32}$ 1,15-diol originates from the low
flow area of freshwater systems (ponds and lakes; Lattaud et al., 2018; Häggi et al., 2019). The combined increase in *n*-alkane concentrations, BIT index, and F$_{C32\ 1,15\text{-diol}}$ points towards increased runoff from the continent. Further evidence for increased terrestrial input comes from a concurrent, rapid decrease in $\delta^{13}C_{bulk}$ values (1.5‰) in parallel with a large increase in C/N ratios (+6‰) and high TOC values (+4%) just after the Ancylus Regression. Altogether, an increase in terrestrial-derived organic matter either from enhanced coastal erosion or riverine input occurred at this time.


### 4.2.2 Ancylus Lake vegetation and hydrological change

Following the Ancylus Regression, there were two shifts in mid- and long-chain *n*-alkane C isotope signatures: an increase (+2‰) around 9.2 ka, followed by a return to lower values, and a second increase (again +2‰) at the onset of the marine transgression (Fig. 4). These shifts in $\delta^{13}C_{alkane}$ values during the AL phase might reflect the advance and retreat of coniferous
cover as conifers, such as *Pinus* and *Juniperus*, have higher $\delta^{13}C_{alkane}$ values compared with angiosperms (Diefendorf and Freimuth, 2017). Since *Pinaceae* produce low concentrations of *n*-alkanes compared to *Juniperus* (Diefendorf et al., 2011; Lane, 2017), these C isotope shifts can be tentatively attributed to *Juniperus* shrub extension. These isotopic shifts align with the main climatic changes recorded in Swedish and Finnish lakes (Seppä and Birks, 2001; Seppä et al., 2005; Antonssson et al., 2006). The maximum extension of *Pinus* and *Juniperus* was recorded at 9.2 ka in these regional lakes (Seppä and Birks,
2001; Seppä et al., 2005; Antonsson and Seppä, 2007), coincident with elevated $\delta^{13}C_{alkane}$ values. Similarly, a large increase in the angiosperm *Alnus* (which tends to have more lower $\delta^{13}C_{alkane}$ values) was observed at 8.7 ka at the time of a decrease in *Juniperus* cover north of the Baltic Sea (Antonsson and Seppä, 2007), paralleling the return to lower $\delta^{13}C$ values.

A significant increase in $\delta^{2}H_{alkane}$ values (~ 50‰) began after 10 ka and peaked at 9.2 ka (Fig. 5), precisely at the time when a similar increase in $\delta^{2}H$ values of haptophyte-derived alkenones from the same core was noted (Weiss et al., 2020). The H
isotope composition of terrestrial plants reflects local meteoric water signals (Sachse et al., 2012), and is thus subject to large fluctuations resulting from variations in the hydrological cycle. H isotope values of precipitation were reconstructed from $C_{23}$ and $C_{29}$ *n*-alkanes using linear models from McFarlin et al. (2019), and range from -125‰ to -62‰ (residual standard error = 14) and -107‰ to -36‰ (residual standard error = 15), respectively, across the AL phase (Fig. 5). The lowest $\delta^{2}H_{alkane}$ values – and in turn $\delta^{2}H$ values of reconstructed precipitation – occur during or immediately preceding the 50‰ increase (Fig. 5).
Reconstructed $^{2}H$ values of precipitation using the $C_{23}$ *n*-alkane are lower than those for the $C_{29}$ *n*-alkane, suggesting an





influence from an isotopically lighter (more negative) water source affected isotopic signatures of aquatically-produced *n*-alkanes. Model data suggests that the H isotope composition of the SIS was around -310‰ (De Boer et al., 2014), and modern mean annual precipitation in the region is -63‰ (Bowen and Revenaugh, 2003), highlighting the marked isotopic offset between the two sources. Larger volumes of meltwater from the SIS would result in significantly more negative $\delta^2$H values

for aquatically-derived mid-chain *n*-alkanes, in line with what is observed here at around 10 ka (Fig. 5) and also recorded by haptophyte algae-derived long-chain alkenones from the same core (Weiss et al., 2020). After 10 ka the decrease of melt water and increase in precipitation (Dahl and Nesje, 1996; Seppä and Birks, 2001; Hammarlund et al., 2003) likely caused the dominant water source to become isotopically heavier, resulting in increased *n*-alkane H isotope values.

The global transition from a glacial into an interglacial climate state across the Holocene, was punctuated by a few abrupt cold

events observed in Northern Hemisphere records (e.g., Fleitmann et al., 2008), and in glacier and ice sheet oscillations. At 9.2 ka, the peak in $\delta^2$H$_{alkane}$ values, there was a slowdown of thermohaline circulation due to a large meltwater pulse from the Laurentide Ice Sheet into the North Atlantic, causing a brief cold event in the Northern Hemisphere (Fleitmann et al., 2008). While our record is of insufficient resolution to capture this rapid event, the increase of $\delta^2$H$_{alkane}$ values noted at 9.2 ka is presumably also influenced by the environmental conditions present at that time. Another rapid cold event occurred at 8.2 ka,

which is not observed in our record, but may be elucidated with higher resolution sampling at this time interval.

### 4.3 Marine Transgression

A large marine transgression re-established the connection between the Baltic Sea and the North Sea at around 7.7 ka (Moros et al., 2002). At the onset of the transgression (which lasted from 7.7 to 7.2 ka), long-chain *n*-alkane (> C$_{27}$) concentrations increased and the shorter-chain *n*-alkanes are detected (< C$_{19}$, Fig. 3), indicating greater input from terrestrial and algal sources,

respectively. Regional warming began just prior to this period (Seppä and Birks, 2001; Seppä et al., 2005; Antonsson and Seppä, 2007), which favored the growth and diversification of terrestrial and aquatic plants. At the start of the transgression, mid- and long-chain *n*-alkane C isotope values began to diverge (LEWIS index increases, Fig. 6b), with the mid-chain *n*-alkanes becoming more positive by about 4‰ (Fig. 4). The $\delta^{13}$C$_{alkane}$ values of long-chain *n*-alkanes showed a C$_3$ higher plant signal (-30.4 ± 0.7‰), while the mid-chain *n*-alkanes were higher (-27.7 ± 0.4‰), characteristic of submerged aquatic plants

(Ficken et al., 2000). The C$_{21}$ *n*-alkane, produced by submerged aquatic plants and freshwater and marine phytoplankton, was the highest prior to the transgression, and became lower by ~5‰ across the transgression, despite the fact that large volumes of saline water entered the basin and the $\delta^{13}$C$_{bulk}$ values showed increased values. Terrestrial vegetation is general relatively $^{13}$C-depleted compared to marine organisms (Fig. 2), so the more negative values noted for the C$_{21}$ *n*-alkane might indicate a relative increase in the abundance of submerged aquatic plants and freshwater phytoplankton rather than the influx of saline

water. An increase in $\delta^2$H$_{alkane}$ values also occurred across the transgression (except the C$_{23}$ *n*-alkane which became more negative). The C$_{23}$ *n*-alkane is known to be produced by *Sphagnum* species (Baas et al., 2000; Vonk and Gustafsson, 2009), and thus may be reflecting a different water source than the rest of the *n*-alkanes. Potential alternative water sources for



*Sphagnum* are the stagnant water of meltwater ponds which would have very negative H isotope values. TOC and $\delta^{13}C_{bulk}$ values increased, while C/N ratios decreased, likely due to enhanced phytoplankton production (cyanobacterial blooms in particular, Bianchi et al., 2000), and amplified input of organic matter into the sediments from increased erosion and primary production (Sohlenius et al., 2001). The BIT index continued to decrease (Fig. 6b), while #ring$_{tetra}$ increased (> 0.7) indicating a change in the main brGDGT producers from (freshwater) lacustrine production towards (marine) sedimentary pore-water production. The decrease in BIT values at the transgression may be the result of a shift in freshwater versus marine producers rather than a decrease in soil-input. A rapid decrease in $F_{C32\ 1,15\text{-diol}}$ occurred from the end of the AL phase into the marine transgression (8.0 to 7.3 ka; Fig. 6b). The decrease may be the result of increased salinity in the basin which likely caused a change in the LCD-producing population from freshwater to brackish and marine LCD-producing species known to synthesize less of the $C_{32}$ 1,15-diol.

### 4.4 Littorina Sea and Modern Baltic phases

The transition from the marine transgression into the early half of the Littorina Sea (LS) phase is characterized by variable $\delta^2H_{alkane}$ and $\delta^{13}C_{alkane}$ values (Figs. 4 and 5). Multiple studies (e.g., Hammarlund et al., 2003; Seppä et al., 2005; Antonsson et al., 2006; Antonsson and Seppä, 2007) indicate that the period between 8.0 and 4.2 ka was warm, dry, and stable, followed by a period of rapid hydrological changes. This is supported by the LDI-reconstructed summer sea surface temperature (Table S1), indicating an optimum between 7.3 and 5.3 ka (LS average of $21.5 \pm 1.5$ °C). In addition, stable #ring$_{tetra}$, BIT, and $F_{C32\ 1,15\text{-diol}}$ suggest relatively unchanged bacterial communities and river inflow. However, the C and H isotope data suggest instability and diversification in the catchment from the marine transgression until the middle of the LS phase. We note, however, that the rapid hydrological changes evidenced in other records and the instability suggested by isotopic data discussed here likely do not capture the nuances of the time period as a result of low sampling resolution. At the transition from the LS into the Modern Baltic phase, the $C_{28}$ 1,14-diol, characteristic of *Proboscia* diatoms (Rampen et al., 2007) and *Apedinella radians* (algae mainly found in estuaries, Rampen et al., 2011) is present in relatively high amounts. This aligns with data from Andrén et al. (2000), which indicated that freshwater diatoms were low throughout the LS phase, but were more prominent during the Modern Baltic phase.

### 5 Conclusions

The Holocene history of the Baltic Sea consisted of variations in regional vegetation and hydrology that were connected to global climate phenomena. The Ancylus Lake phase, in particular, was characterized by large fluctuations in the extent of gymnosperm cover correlated with the gradual warming in the catchment and substantial shifts in source water from a significantly light, meltwater-influenced source to an isotopically heavier precipitation-dominated source. These findings align with a previous study which found a similar source water shift over the same time interval. Subsequent large meltwater pulses

from the Laurentide Ice Sheet, which caused a slowdown of thermohaline circulation that triggered two rapid, regional cold

events at 9.2 and 8.2 ka, likely contributed to the lack of diversification of terrestrial vegetation noted for this period. The marine transgression that ended the low salinity Ancylus Lake phase was accompanied by a diversification of terrestrial higher plants as well as changes in coastal aquatic ecosystems as a result of regional warming which continued into the Late Holocene. Our record showed an offset between previously established phase boundaries based on salinity and eustatic changes and the fluctuations in hydrology and higher plant diversification. Proxy signals like the record discussed here highlight the complex

temporal dynamics at play between paleo archives. For example, shoreline changes can be temporally offset from riverine transport of plant waxes. Use of multiple proxies is essential to fully understand the complexity of paleoenvironments, but spatial and temporal resolution of individual proxies must be acknowledged.

**Author contribution**

GW and JL designed the study, extracted samples, conducted isotopic analyses, and contributed equally to writing of the

manuscript with contributions from all co-authors.

**Competing interests**

The authors declare that they have no competing interests.

**Acknowledgments**

We thank Jort Ossebaar and Ronald van Bommel for support with EA-IRMS, GC-MS and GC-IRMS; Denise Dorhout is

thanked for helping with UHPLC analyses. The compound-specific carbon isotope ratios were analyzed at the Climate Geology group of ETH Zurich with help from Steward Bishop. J.L. is funded by a Rubicon grant (019.183EN.002) from the Netherlands Organization for scientific research (NWO). All acquired data will be stored in the PANGAEA database, which can be accessed at doi:

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






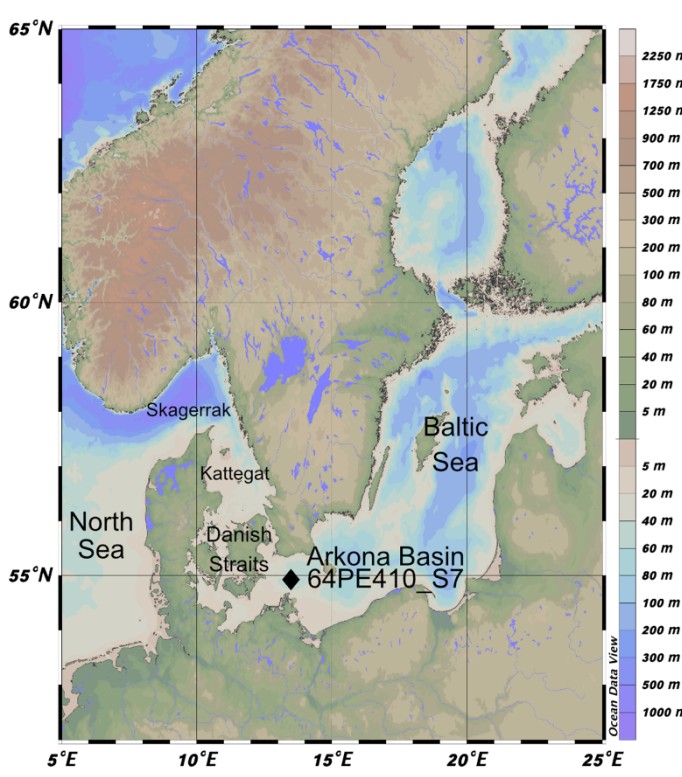

**Figure 1: Location of 64PE410_S7 in the Southwestern Baltic Sea near the connection with the North Sea via the Danish Straits. Map was created in Ocean Data View (Schlitzer, R., Ocean Data View, https://odv.awi.de, 2018).**





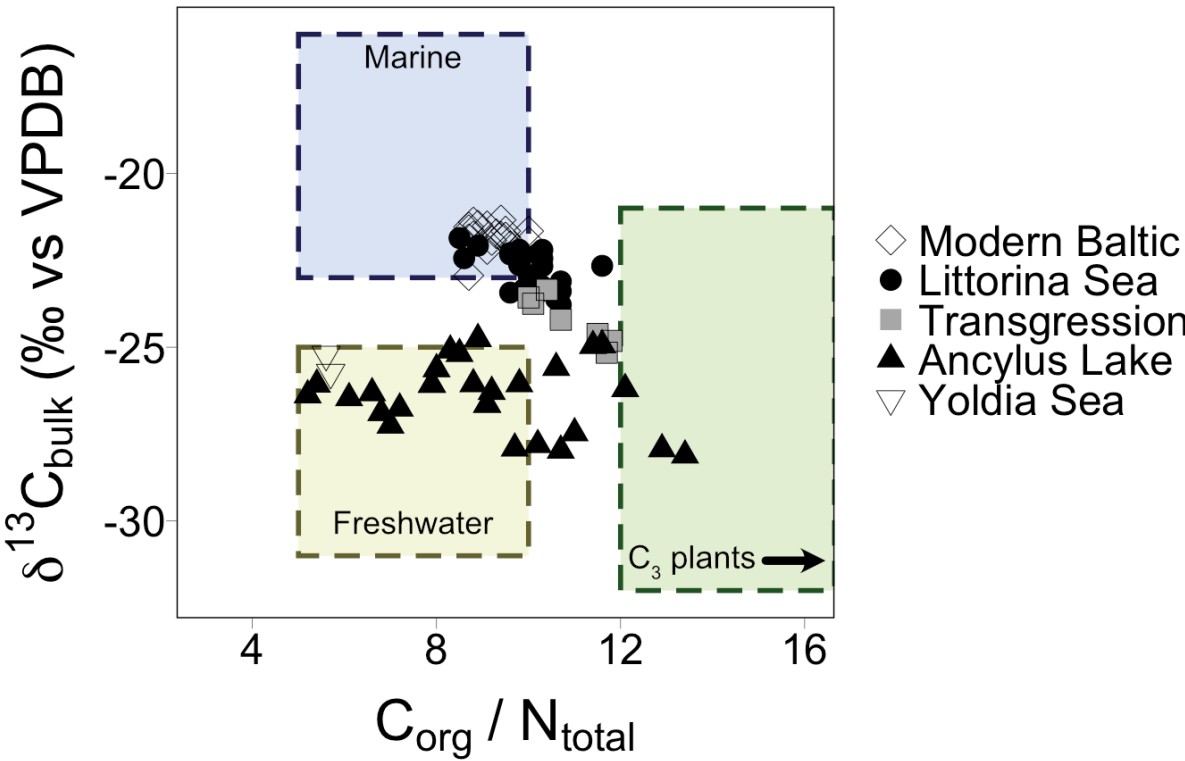


**Figure 2: Elemental C/N ratios plotted against $\delta^{13}C$ values of bulk sediment. Shapes correspond to different Baltic Sea phases and align with a mix of values corresponding to marine, freshwater, and $C_3$ plant signatures. Arrow in the $C_3$ plants box indicates that values for $C_3$ plants encompass a wider range of values which are excluded from this plot for clarity.**



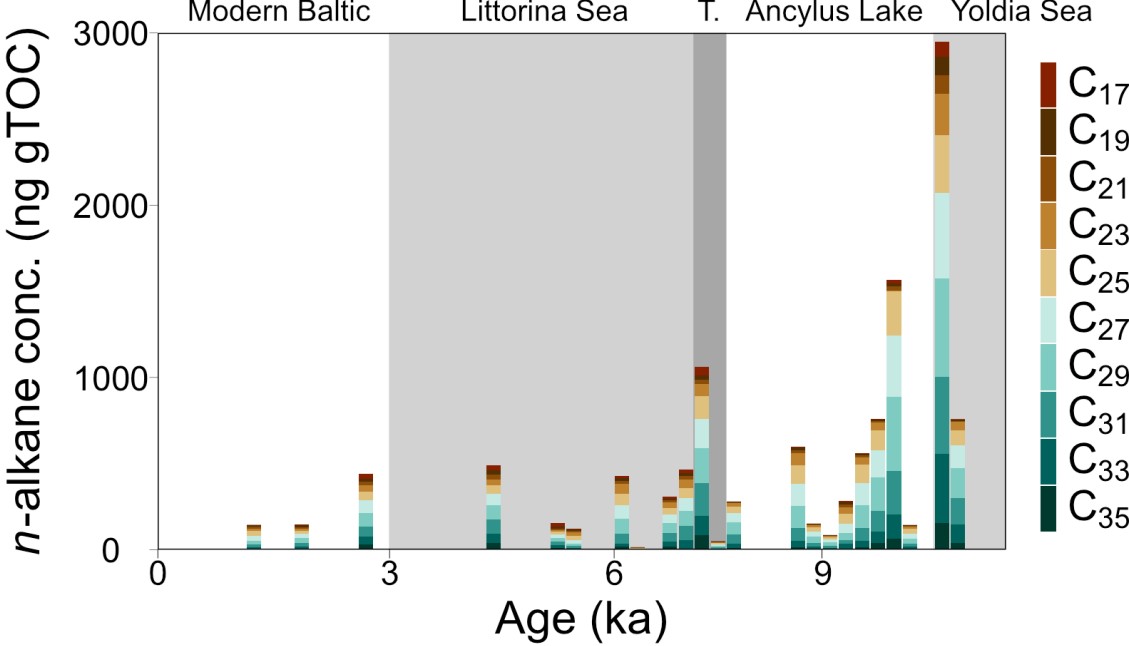


**Figure 3: TOC weighted concentrations of $C_{17} - C_{35}$ *n*-alkanes in core 64PE410_S7. Concentrations were highest in the Yoldia Sea brackish phase and lowest in the Modern Baltic.**


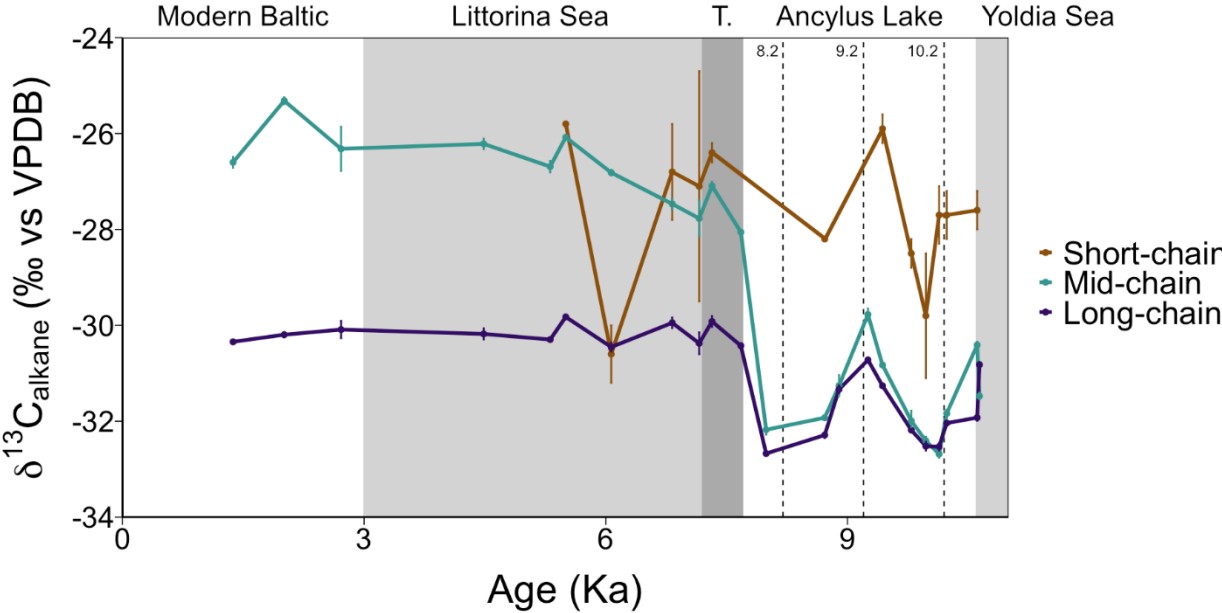

**Figure 4: Carbon isotope ratios of short-chain (C$_{21}$), mid-chain (C$_{23}$ – C$_{25}$), and long-chain (C$_{27}$ – C$_{31}$) *n*-alkanes from core 64PE410_S7. Phase boundaries are indicated (T stands for marine transgression) as well as a regional climate event at 10.2 ka (Ancylus Regression), and large-scale (northern hemisphere) cold events at 9.2 ka and 8.2 ka.**





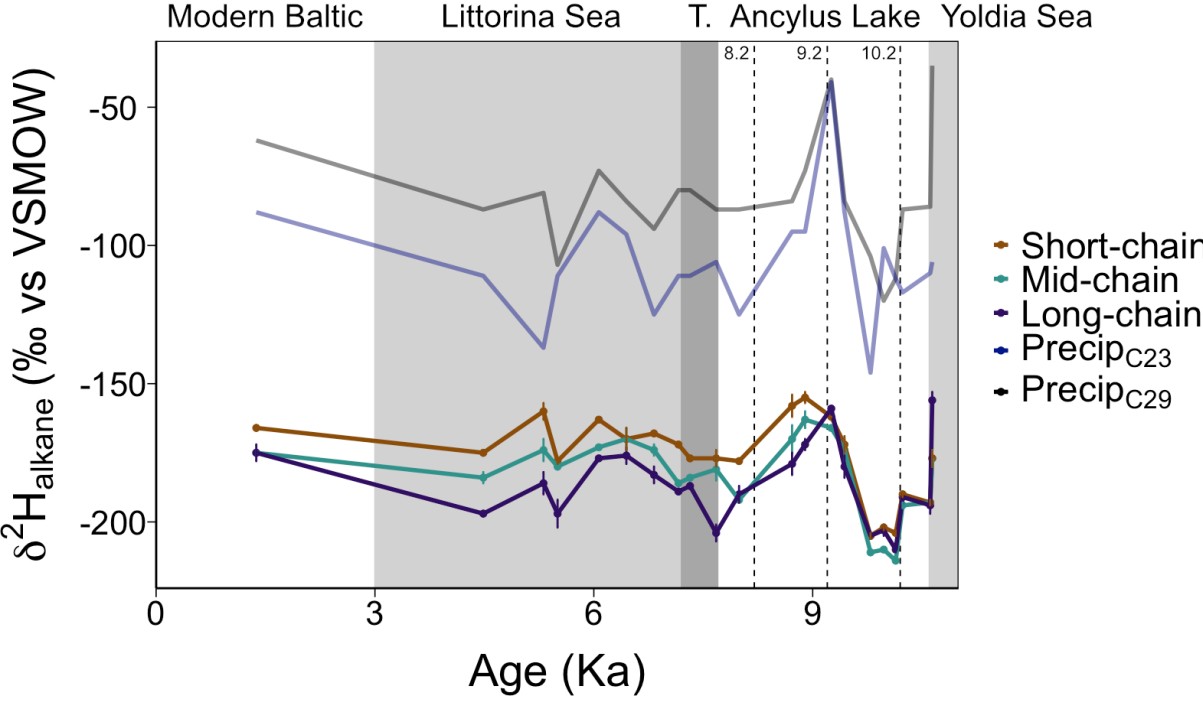

**Figure 5: Hydrogen isotope ratios of short-chain, mid-chain, and long-chain $C_{21} - C_{33}$ $n$-alkanes from core 64PE410_S7. Phase boundaries are indicated (T stands for marine transgression) as well as a regional climate event at 10.2 ka (Ancylus Regression), and larger scale events at 9.2 ka and 8.2 ka. Blue and black lines represent reconstructed $\delta^2H$ values of precipitation from linear models of McFarlin et al. (2019) using $\delta^2H$ values of $C_{23}$ and $C_{29}$ $n$-alkanes.**







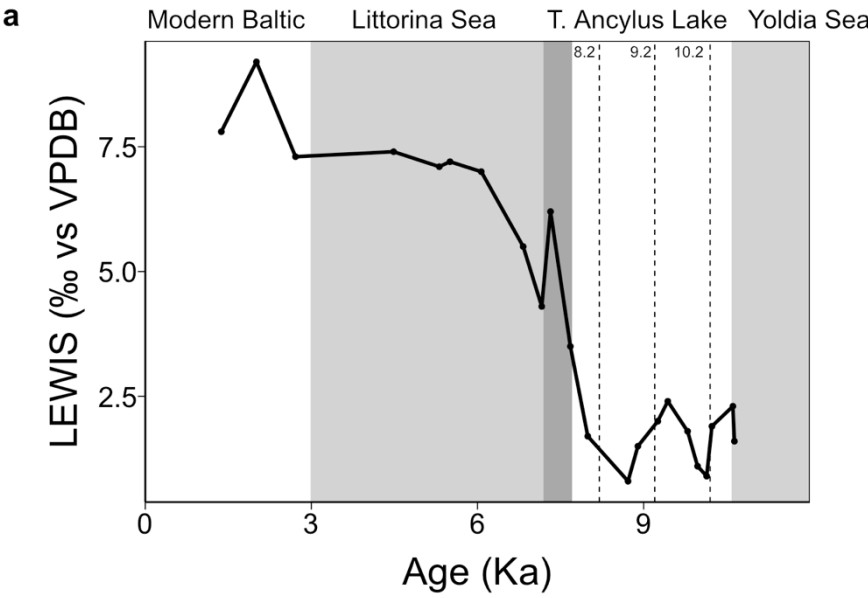

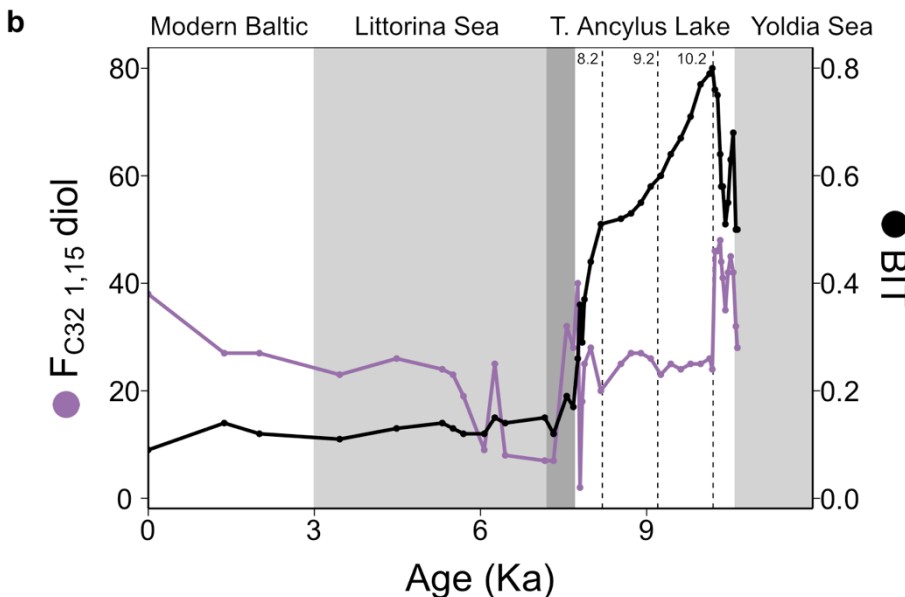

**Figure 6: (a) The LEaf-Wax Isotopic Spread (LEWIS) index (Magill et al., 2019) shows a marked increase from the end of the Ancylus Lake phase into the Modern Baltic indicating increased species diversity in the younger half of the record. (b) Fractional abundance of $C_{32}$ 1,15-diol ($F_{C32\ 1,15\text{-diol}}$, de Bar et al., 2016; Lattaud et al., 2017) and BIT index (Hopmans et al., 2004) for core**

**64PE410_S7, indicators of riverine and terrestrial runoff into the basin, suggest the highest terrestrial input occurred during the Ancylus Lake phase and decreased thereafter. Phase boundaries are indicated (T stands for marine transgression) as well as a regional climate event at 10.2 ka (Ancylus Regression), and larger scale events at 9.2 ka and 8.2 ka.**



**Table 1: Summary of the proxies applied in this study, how they are used as paleoclimate indicators, and their source(s).**


| Organic compound | Uses | Source | References |
|---|---|---|---|
| Bulk C and N isotopes | Sources of organic matter | Organic matter | Meyers, 1994; Leng and Lewis, 2017 |
| Short-chain $n$-alkanes ($C_{17}$ - $C_{21}$) | Reconstruction of vegetation (concentration and C isotopes); growth water, hydrology (H isotopes) | Phytoplankton, bacteria | Meyers and Ishiwatari, 1993; Zhang and Sachs, 2007; Sachse et al., 2012; |
| Mid-chain $n$-alkanes ($C_{23}$ - $C_{25}$) | Reconstruction of vegetation (concentration and C isotopes); growth water, precipitation (H isotopes) | Submerged macrophytes | Ficken et al., 2000; Sachse et al., 2012 |
| Long-chain $n$-alkanes (> $C_{27}$) | Reconstruction of vegetation (concentration and C isotopes); growth water, precipitation (H isotopes) | Terrestrial higher plants | Eglinton and Hamilton, 1967; Sachse et al., 2012; Freimuth et al., 2017 |
| BIT index | Aquatic production and soil erosion | Heterotrophic bacteria | Hopmans et al., 2004; Weijers et al., 2007, 2009 |
| $C_{32}$ 1,15 diol | Riverine input | Marine and freshwater Eustigmatophytes | Volkman et al., 1992; Rampen et al., 2012; de Bar et al., 2016; Lattaud et al., 2017; Balzano et al., 2018; |
| LDI | Sea surface temperature | Marine and freshwater Eustigmatophytes | Rampen et al., 2012; de Bar et al., 2020 |
| #ring$_{tetra}$ | Sediment pore-water branched glycerol dialkyl glycerol tetraether (brGDGT) production | Heterotrophic bacteria | Sinninghe Damsté, 2016 |





**Table 2: Carbon isotope ratios of $C_{21} - C_{31}$ $n$-alkanes from core 64PE410_S7, n.m. denotes when concentrations were too low for duplicate isotope measurements.**


| Age (kyr) | Depth (cm) | δ¹³C$_{C21}$ (‰ vs. VPDB) | S.D. | δ¹³C$_{C23}$ (‰ vs. VPDB) | S.D. | δ¹³C$_{C25}$ (‰ vs. VPDB) | S.D. | δ¹³C$_{C27}$ (‰ vs. VPDB) | S.D. | δ¹³C$_{C29}$ (‰ vs. VPDB) | S.D. | δ¹³C$_{C31}$ (‰ vs. VPDB) | S.D. |
|---|---|---|---|---|---|---|---|---|---|---|---|---|---|
| 1.37 | 120 | n.m. | n.m. | -24.6 | 0.2 | -27.9 | 0.2 | -29.2 | 0.1 | -30.5 | 0.0 | -32.4 | 0.1 |
| 2.01 | 220 | n.m. | n.m. | -23.2 | 0.2 | -26.7 | 0.1 | -28.0 | 0.3 | -30.5 | 0.2 | -32.3 | 0.2 |
| 2.71 | 320 | -23.4 | 1.6 | -24.7 | 1.7 | -27.5 | 0.3 | -28.5 | 1.1 | -30.0 | 0.0 | -32.0 | 0.0 |
| 4.49 | 520 | -22.6 | 0.9 | -24.6 | 0.4 | -27.2 | 0.1 | -28.3 | 0.1 | -29.9 | 0.0 | -32.0 | 1.0 |
| 5.31 | 600 | -24.7 | 1.0 | -26.3 | 0.4 | -28.3 | 0.1 | -29.5 | 0.1 | -31.1 | 0.2 | -31.7 | 0.1 |
| 5.50 | 620 | -25.8 | 0.0 | -24.3 | 0.0 | -27.7 | 0.0 | -28.4 | 0.0 | -29.8 | 0.0 | -31.5 | 0.1 |
| 6.07 | 680 | -30.6 | 0.5 | -25.6 | 0.2 | -27.9 | 0.0 | -28.9 | 0.2 | -30.4 | 0.2 | -32.6 | 0.0 |
| 6.83 | 760 | -26.8 | 0.0 | -25.9 | 0.4 | -29.0 | 0.4 | -28.8 | 0.0 | -29.7 | 0.1 | -31.4 | 0.2 |
| 7.16 | 800 | -27.1 | 1.7 | -27.7 | 1.1 | -28.5 | 0.2 | -29.7 | 0.3 | -30.9 | 1.2 | -31.5 | 0.0 |
| 7.32 | 840 | -26.4 | 0.2 | -25.3 | 0.2 | -28.1 | 0.2 | -28.5 | 0.1 | -29.7 | 0.1 | -31.5 | 0.0 |
| 7.68 | 930 | -21.4 | 0.4 | -27.3 | 0.0 | -28.7 | 0.0 | -29.5 | 0.2 | -31.0 | 0.1 | -30.8 | 0.2 |
| 7.99 | 990 | n.m. | n.m. | -32.4 | 0.1 | -31.5 | 0.6 | -31.9 | 0.2 | -32.3 | 0.2 | -34.1 | 0.1 |
| 8.72 | 1030 | -28.2 | 0.0 | -32.3 | 0.2 | -31.7 | 0.0 | -31.9 | 0.0 | -32.2 | 0.3 | -33.1 | 0.4 |
| 8.90 | 1040 | n.m. | n.m. | -31.9 | 0.7 | -30.9 | 0.2 | -30.4 | 0.4 | -31.8 | 0.2 | -31.9 | 0.4 |
| 9.26 | 1060 | n.m. | n.m. | -29.5 | 0.0 | -29.9 | 0.3 | -30.1 | 0.3 | -30.7 | 0.1 | -31.6 | 0.1 |
| 9.44 | 1070 | -25.9 | 0.2 | -29.9 | 0.1 | -31.5 | 0.1 | -30.4 | 0.0 | -31.8 | 0.2 | -32.2 | 0.2 |
| 9.79 | 1090 | -28.5 | 0.2 | -31.9 | 0.4 | -32.0 | 0.4 | -31.5 | 0.0 | -32.2 | 0.0 | -33.3 | 0.3 |
| 9.97 | 1100 | -29.8 | 1.0 | -32.2 | 0.3 | -32.5 | 0.1 | -32.0 | 0.2 | -32.6 | 0.1 | -31.6 | n.m. |
| 10.14 | 1110 | -27.7 | 0.5 | -32.4 | 0.0 | -32.7 | 0.1 | -32.0 | 0.0 | -32.5 | 0.1 | -33.3 | 0.0 |
| 10.23 | 1130 | -27.7 | 0.3 | -31.7 | 0.3 | -31.9 | 0.1 | -31.2 | 0.0 | -32.2 | 0.2 | -33.1 | 0.0 |
| 10.61 | 1210 | -27.6 | 0.3 | -30.1 | 0.0 | -30.6 | 0.2 | -31.3 | 0.0 | -32.1 | 0.4 | -32.4 | 0.0 |
| 10.64 | 1216 | n.m. | n.m. | -31.7 | 0.0 | -31.4 | 0.2 | -30.2 | 0.1 | -30.5 | 0.1 | -31.7 | 0.3 |





**Table 3: Hydrogen isotope ratios of $C_{21} - C_{33}$ *n*-alkanes from core 64PE410_S7, n.m. denotes when concentrations**
**were too low for duplicate isotope measurements.**

| Age (kyr) | Depth (cm) | $\delta^2H_{C21}$ (‰ vs. VSMOW) | S.D. | $\delta^2H_{C23}$ (‰ vs. VSMOW) | S.D. | $\delta^2H_{C25}$ (‰ vs. VSMOW) | S.D. | $\delta^2H_{C27}$ (‰ vs. VSMOW) | S.D. | $\delta^2H_{C29}$ (‰ vs. VSMOW) | S.D. | $\delta^2H_{C31}$ (‰ vs. VSMOW) | S.D. | $\delta^2H_{C33}$ (‰ vs. VSMOW) | S.D. |
|---|---|---|---|---|---|---|---|---|---|---|---|---|---|---|---|
| 1.37 | 120 | -158 | 2 | -172 | 2 | -176 | 2 | -174 | 6 | -174 | 7 | -179 | 5 | n.m. | n.m. |
| 4.49 | 520 | -167 | 1 | -183 | 1 | -185 | 3 | -189 | 1 | -196 | 0 | -200 | 0 | -204 | 5 |
| 5.31 | 600 | -146 | 3 | -174 | 4 | -175 | 8 | -174 | 6 | -186 | 8 | -194 | 9 | -188 | 5 |
| 5.50 | 620 | -171 | 1 | -181 | 3 | -179 | 1 | -182 | 6 | -194 | 12 | -204 | 2 | n.m. | n.m. |
| 6.07 | 680 | -132 | 0 | -172 | 1 | -173 | 0 | -173 | 0 | -179 | 1 | -181 | 1 | n.m. | n.m. |
| 6.45 | 720 | -158 | 3 | -174 | 7 | -168 | 3 | -164 | 4 | -180 | 4 | -174 | 6 | -198 | 10 |
| 6.83 | 760 | -134 | 2 | -178 | 1 | -170 | 4 | -177 | 1 | -184 | 8 | -182 | 9 | -190 | 2 |
| 7.16 | 800 | -141 | 1 | -183 | 1 | -188 | 1 | -187 | 1 | -190 | 2 | -192 | 0 | -185 | 4 |
| 7.32 | 840 | -162 | 1 | -182 | 1 | -186 | 0 | -183 | 0 | -190 | 2 | -188 | 2 | -187 | 3 |
| 7.68 | 930 | -196 | 3 | -174 | 6 | -187 | 6 | -189 | 6 | -197 | 10 | -220 | 0 | -221 | 0 |
| 7.99 | 990 | -177 | 1 | -178 | 1 | -199 | 0 | -193 | 1 | -196 | 1 | -181 | 10 | -183 | 1 |
| 8.72 | 1030 | -158 | 6 | -158 | 7 | -178 | 8 | -178 | 0 | -181 | 0 | -175 | 0 | n.m. | n.m. |
| 8.90 | 1040 | -157 | 1 | -154 | 4 | -169 | 5 | -169 | 7 | -172 | 4 | -179 | 2 | -168 | 0 |
| 9.26 | 1060 | -162 | 1 | -162 | 0 | -168 | 1 | -158 | 1 | -157 | 1 | -162 | 1 | n.m. | n.m. |
| 9.44 | 1070 | -174 | 4 | -172 | 4 | -176 | 7 | -174 | 4 | -182 | 10 | -189 | 12 | n.m. | n.m. |
| 9.79 | 1090 | -196 | 1 | -207 | 2 | -213 | 2 | -207 | 1 | -206 | 2 | -199 | 4 | n.m. | n.m. |
| 9.97 | 1100 | -196 | 1 | -203 | 0 | -213 | 1 | -207 | 0 | -204 | 4 | -196 | 2 | -199 | 5 |
| 10.14 | 1110 | -203 | 2 | -207 | 1 | -214 | 1 | -211 | 0 | -212 | 0 | -207 | 2 | -209 | 1 |
| 10.23 | 1130 | -186 | 2 | -191 | 2 | -195 | 0 | -190 | 4 | -196 | 1 | -185 | 3 | n.m. | n.m. |
| 10.61 | 1210 | -203 | 1 | -187 | 2 | -197 | 0 | -192 | 3 | -197 | 2 | -198 | 4 | -190 | 12 |
| 10.64 | 1216 | -193 | 2 | -175 | 5 | -178 | 2 | -159 | 6 | -149 | 3 | -162 | 11 | -151 | 3 |