# Peer review of "Co-evolution of terrestrial and aquatic ecosystem structure with hydrological change in the Holocene Baltic Sea"

_Climate of the Past, 2020_

## Author Response (AR1)

Dear Editor-in-Chief,

Please find resubmitted the manuscript entitled "Co-evolution of the terrestrial and aquatic ecosystem in the Holocene Baltic Sea". Note the change of title as suggested by the reviewers. We thank you, and the reviewers, for your positive comments and we have used these to revise and improve our manuscript. Below we have listed our responses (in bold) to the reviewers concerns and suggested changes. In the revised manuscript the changes are highlighted in yellow. We have added a discussion of regional vegetation from nearby lacustrine palynological records and examined how our biomarker isotopic evidence aligns with regional trends. We discuss data by Baltic Sea phase and propose a new division of the Ancylus Lake phase based on the shift in source water input from melt water of the Scandinavian Ice Sheet to primarily precipitation that occurs mid-phase when $n$-alkane distributions are otherwise stable. We hope that we have addressed all reviewers comments with this revision.

On behalf of the co-authors,

Gabriella M. Weiss

**Response to reviewer 1**

This is a well-written manuscript that presents analyses of biomarkers and stable isotopes of specific compounds. The analyses are state of the art and have not previously been used in the south-western Baltic. It is important to use new methods to improve our understanding of past environmental changes. However, the interpretations of the data are not straightforward and after reading the manuscript I did not feel that I learned much new about the history of the Baltic Sea. I am sorry but I don't feel that the study "provided new insights into changes … in the vegetation in the western Baltic region throughout the Holocene" (line 210).

**We thank the reviewer for their praise and constructive comments on our dataset. We have restructured the manuscript to focus more on the hydrological shifts noted in the isotope data and how our results confirm diversification of vegetation in the region after ~10.5 ka (see 4.2.1 and 4.2.2).**

The interpretations are hampered by the fact that the geography of the Arkona Basin changed over time. During the Yoldia Sea stage it was a bay of the Baltic Basin, which was connected to the Kattegat via straits in south-central Sweden. During the early part of the Ancylus Lake stage the outflow shifted from south-central Sweden to what became the Danish straits. During the Littorina Sea stage it was part of the Baltic Sea, with outflowing and inflowing water masses. Water level varied a lot over time due to glacioisostatic rebound and global eustatic sea level rise.

**We agree that the geography of the basin changed a lot during the Holocene. Our main interest lies in the terrestrial catchment and linking plant biomarker isotopes to changes in vegetation and regional hydrology. The main impact of the geographical changes occurring in the basin will be on the distance from the continent to the core site (i.e., decreasing with decreased sea level). In addition, increased sea level can cause enhanced coastal erosion, mobilizing soil and plant matter. We have added more discussion on the impact of sea-level changes in the revised manuscript.**

Another issue concerns erosion, reworking and redeposition. Erosion is particularly important at the transition from the Ancylus Lake to the Littorina Sea. Could it be that reworking explains the low $\delta^{13}C$ values of bulk sediment samples from the Transgression phase?

**During the transgression there is a shift from low $\delta^{13}C_{bulk}$ (characteristic of $C_3$ plants and freshwater algae, Fig. 2) to higher values typical for marine environments. It is possible that, at the beginning of the transgression, enhanced erosion of coastal soils explains the sharp peak in TOC (Fig. S1) although it is not reflected in the $\delta^{13}C_{bulk}$ which seem to be recording the shift from freshwater to a more marine environment.**

It is mentioned that some of the molecules analyzed in the study can be wind transported (line 69). I wonder if the molecules in the sediment formed in the Arkona Basin and the surrounding land area? Or could it be that the molecules come from the Kattegat and were transported by inflowing water to the Arkona Basin? Or did they come from northern parts of the Baltic Sea or from the whole drainage area of the Baltic Sea?

**Long-chain alkanes were likely produced by vegetation in the whole drainage area of the Baltic Sea. The short-chain alkanes could have been produced in the Arkona basin or transported into the basin from surrounding lakes and rivers.**

The authors discuss the results "by Baltic Sea phase"(s) (line 212). The phases are Yoldia Sea, Ancylus Lake and Littorina Sea. However, the discussion is divided into: Yoldia Sea, Ancylus Lake,

Ancylus Regression, Ancylus Lake vegetation and hydrological change, Marine transgression and finally Littorina Sea and Modern Baltic phases. I suggest that the authors use only three headings. What they term Marine Transgression is part of the Littorina Sea stage.

**We changed the division of the discussion in the revised manuscript to follow this suggestion (4.2 Ancylus Lake is now divided into 4.2.1 Hydrology and 4.2.2 Vegetation).**

Modern palaeostudies relies on well-dated records and high temporal resolution. However, I get the feeling that the chronology of the studied record is poorly constrained. The age-depth model needs to be described in detail in the paper. It is mentioned that the age-depth model is described in detail in Weiss et al. (2020) and that is was created by combining $^{14}$C-ages of mollusk shells and … (line 99). According to Weiss et al. (2020) three radiocarbon ages were obtained. I could find no information on the ages or which species was used for dating in the 2020 paper or in the supplementary material. Also, I saw no information about which calibration curve was used. It should also be explained in the paper that only calibrated ages are discussed. In particular, I wonder how the older non-marine part of the core was dated.

**The core was dated by correlating XRF data to two nearby cores described in Warden et al. (2016). The $^{14}$C ages were corrected using local marine reservoir values from Lougheed et al. (2013; doi:10.5194/cp-9-1015-2013). We added a more detailed discussion of this information to the revised manuscript (L130-150), which now reads:  The geochemical composition of the core was analyzed by X-ray fluorescence (XRF) using an Avaatech XRF core scanner with a 100W rhodium X-ray tube and a Rayspec cubed SiriusSD silicon drift detector. Continuous measurement of core sections was conducted at 1 cm resolution. The spectral data was processed using bAxil spectrum analysis software to determine the element intensities in counts.**

**The age model was created by combining $^{14}$C-ages of mollusk shells and correlation of Ca/Ti and Br records with two nearby cores (318310: 54°50.34' N; 13°32.03' E, and 318340: 54°55.77' N; 13°41.44' E , Warden et al., 2016, Fig. 1). Bromine, a good indicator of marine organic carbon (Ziegler et al., 2008) was correlated with the total organic carbon content determined by loss on ignition (LOI) in core 318310 (Warden et al., 2016). Five shifts in Arkona Basin sediments are clearly distinguishable in the carbonate (Ca/Ti) and organic matter (Br and LOI) contents of the two sediment cores from Warden et al. (2016) and described in detail in Weiss et al. (2020). Three mollusk shells were radiocarbon dated by Beta Analytic to improve age control for the interval younger than 7 ka. Radiocarbon ages were corrected for the local reservoir effects (reservoir age = 376 years, Lougheed et al., 2013) and calibrated using CALIB (Stuiver et al., 2018). The final age model for all depths was created using the Bacon software package in R (Blaauw & Christen, 2011). Error estimates for $^{14}$C ages were also used for the XRF tie-points (as these are based on correlation to the nearby $^{14}$C dated cores). In Bacon, student's t-test values were set to 33 and 34 to allow for more narrow error estimates as suggested in the Bacon manual.**

With respect to resolution, I note that four samples were analysed for the time period from 6 ka to the present. The resolution is higher in the older part of the record.

**The focus of our study is on the Ancylus Lake, hence the difference in sampling resolution. We have made this more explicit in the revised version of the manuscript (L159-160: "Briefly, sediments were sampled every 100 cm at the top of the core and every 10 cm after 930 cm to better understand the climatic events in the Early Holocene. Freeze-dried sediment samples (11 – 17 grams) were extracted via an accelerated solvent extractor").**

The paper must also provide information about the sediments in the core, based on a visual core description.

**As noted above, the core description comes from the geochemistry of the core, which was determined by X-ray fluorescence (XRF) analysis. XRF data was used for the age model which is published by Weiss et al. (2020).**

One of the main scientific questions in the development of the Arkona Basin and the Baltic Basin is the dating of the transition from the Ancylus Lake to the Littorina Sea. Andrén et al . (2000) dated it to ca. 10.1 cal. ka BP and Berglund et al. (2005) dated it to ca. 9.8 cal. ka BP, whereas other studies have dated it to 7-8 cal. ka BP. I am surprised to see that this question is not mentioned in the manuscript. Did the study provide any information on this issue?

**Based on the age model (discussed in section 2), we suggest a later transition from the Ancylus Lake into the Littorina Sea. This is based on an increase in organic matter content at 7.7 ka and a carbonate maximum (signalling the marine transgression maximum) at 7.2 ka.**

I think one of the most interesting outcomes of the study it that the Ancylus Lake stage was highly dynamic with large isotopic shifts. But the interpretation is difficult. It was partly caused by decreasing influence of melt water. But I suggest that the shift in drainage may also play a role. I am not sure about the importance of the short-lived cold events during this time.

**We agree with the reviewer that the Ancylus Lake phase was dynamic. We added more discussion about the shift in drainage as another possible contributing factor to the complexity of the time in the revised version (L313-330). However, we note that such depleted hydrogen isotope values of lipids followed by a large, positive shift is likely not defined by drainage alone, but rather changes in the relative amount of melt water versus precipitation serving as source water for *n*-alkane-producing organisms.**

Other comments
Line 15. According to the first sentence in the abstract: "The Baltic Sea experienced a number of marine transgressions and regressions throughout the Holocene". However, only one marine transgression is discussed in the manuscript. The northern part of the Baltic Sea experienced regression throughout the Holocene, whereas the southern part experienced one marine transgression during the Holocene.

**We revised this statement to (L 15-16): "The Baltic Sea experienced changes in marine input throughout the Holocene as substantial regional ice melt led to isostatic adjustment and periodic isolation from the North Sea."**

Line 16. According to the next sentence: "These fluctuations in sea level coupled with substantial regional ice melt led to isostatic adjustment and periodic isolation from the North Sea". Isostatic adjustment was caused by ice retreat, not by the "fluctuations in sea level".

**Thank you for catching this. We have removed this sentence.**

Line 28-29. "that promote … diverse phytoplankton communities". A reference is needed. Is the diversity larger than in other areas?

**We added the following references to the end of this sentence (L30): e.g., Wasmund and Uhlig (2003) doi: 10.1016/S1054-3139(02)00280-1; Golubkov et al. (2020), doi: 10.1016/j.oceano.2020.11.002.**

Line 31, 33. Retreat was caused by melting and calving.

**We added "calving" to the reason for the retreat of the SIS (L42).**

Line 33. "the Scandinavian Ice Sheet covered large swaths of Europe". Change to: the Scandinavian Ice Sheet covered large parts of northern Europe.

**We deleted this sentence from the revised manuscript.**

Line 35. The Yoldia Sea stage began in the earliest part of the Holocene. Therefore it was not caused by global and regional temperatures that continued to rise during the Holocene.

**We removed "during the Holocene" in the revised version (L39-41: "As global and regional temperatures fluctuated at the start of the Holocene, rising water levels led to an inundation of saline North Sea water into the lake, resulting in the transition of the basin from a freshwater to a brackish lake, a phase known as the Yoldia Sea (YS) (11.2 to 10.6 ka).").**

Line 38. Sea level fluctuations. It should be water level fluctuations, because lake stages are also involved.

**This is corrected in the revised manuscript (L39-40: "As global and regional temperatures fluctuated at the start of the Holocene, rising water levels…").**

Line 38. The Ancylus Lake was a freshwater lake - I find it a bit strange to call it a low salinity phase.

**We respectfully disagree. Some brackish diatom species have been found in Ancylus Lake sediments (Alhonen, 1972) and Winterhalter (1992) noted that saline water flowed into the basin during the Ancylus Lake phase. It seems most likely that the surface water layer was fresh, but there was saline water at depth.**

Line 41. The freshening was probably caused by land uplift, increased precipitation and decreased evaporation, not by lack of large marine transgressions.

**We omitted "lack of large marine transgression" in the revised manuscript.**

Line 42. until now the authors have discussed salinity changes, but now they state that "The complex climate dynamics caused substantial shifts in the salinity of the Baltic Sea during the Holocene, indicated by changes in the phytoplankton population". However, the main salinity changes were not caused by climatic changes.

**We respectfully disagree and think the main salinity changes are linked with climatic changes. The melting of the SIS followed by large freshwater input and isostatic changes are all linked to climate, directly or indirectly. These factors contributed to the changes in the regional environment (i.e., salinity of the basin) which in turn led to shifts in the phytoplankton (and higher plant) populations.**

Line 57. recalcitrant – is that the same as resistant?

**Yes.**

Line 60, 61. Are there any relevant C4 and CAM plants in the region?

**To the best of our knowledge there are no $C_4$ or CAM plants in the region.**

Line 71. It is mentioned that wind-transported *n*-alkanes are generally deposited within weeks. Does this mean that part of the *n*-alkanes could have their origin in North America?

**Alkanes thought to be from the Sahara desert were found in Atlantic sediments, but in low abundance (Schreuder et al., 2018, 10.1016/j.orggeochem.2017.10.010). With Westerlies being the prevailing winds, it seems unlikely that *n*-alkanes in our record were transported all the way from North America.**

Line 74. preserve information – change to can provide information.

**We changed in the revised manuscript (L73: "organic geochemical tools that provide a wealth of paleoclimatic information".).**

Line 77. convolutes – is this the correct word?

**We changed this to "complicates" to avoid confusion (L107).**

Line 96. what was the diameter of the piston core?

**The piston core has a diameter of 10 cm (L127 in the revised manuscript).**

Line 97. what was the water depth at the core site?

**The water depth at the core site was around 45 m (L129 in the revised manuscript).**

Line 98. Arkona Basin not Arkona basin.

**We changed into "Basin" in the revised manuscript.**

Line 110. I don't think the authors extracted sediment samples, it should be lipids.

**Organics were extracted from the sediments. We changed the sentence in the revised manuscript into (L159): "Samples for lipid analyses…".**

Line 161. The authors note large difference between the piston core described in the manuscript and nearby cores. I wonder if these differences could be explained if the other cores are gravity cores?

**It seems our wording was not clear here. The nearby cores were very similar to the one studied here and also recorded large changes in sedimentation rate. We have changed the wording in the revised manuscript (L216-217) to: "The large variations in sedimentation rates for core 64PE410-S7, and the nearby cores to which it has been correlated …".**

Line 199. The temperature reconstructions for the early part of the record is not similar to those of Kotthoff et al. (2017). The record by Kotthoff only went back to ca. 7.4 ka. I don't think that 18.5°C for the Yoldia Sea phase, 11.5°C for the Ancylus Lake phase and 24° fit with other temperature reconstructions from the region.

**It is true that the record of Kothoff et al. (2017) only goes back to 7.4 ka. As the only other record of temperatures reconstructed using LDI in the area, we felt it should be included. Our record has similar reconstructed temperatures as presented in Kothoff et al. (2017). Because our main focus was not on temperature, we originally chose to omit more detailed information about this to avoid complicating the discussion. We added more information about the utility of**

this proxy in the Baltic and the similarity with other Baltic temperature reconstructions in the revised version (L256-260, 282-283, 314-316, 409-411, 418-424).

Line 212. change Baltic Sea phase to Baltic Sea phases.

**We have changed this in the revised manuscript.**

Line 215. The salinity of the Yoldia Sea phase was first discussed from the presence of *Yoldia arctica* (now *Portlandia arctica*, a marine bivalve). However, marine species are only recorded from the Baltic proper (including the Gotland Deep discussed by Sohlenius et al.). Not sure if any marine species have been recorded from the southern part of the Baltic Basin.

**Alkenone distributions characteristic of marine alkenone producing species were observed in these samples (described in Weiss et al., 2020).**

Line 218. Yoldia Sea sediments in the Arkona Basin are usually considered to be barren of fossils (except for reworked pollen and spores). It is interesting to see that the authors found diols that are produced by freshwater eustigmatophytes. However, I wonder if these algae lived in the Arkona Basin, or in rivers and pools in the catchment?

**Eustigmatophytes have soft tissues only, but the organic compounds they produce (i.e., diols) do preserve well in sediments and are colloquially known as "chemical fossils". It is likely that eustigmatophytes producing C32 1,15-diol were living in the rivers and ponds of the catchment as suggested by previous studies (Lattaud et al., 2017). The producers of other diols (C30 1,15-diol for example) might live in the Arkona Basin.**

Line 234. The pollen records referred to are not nearby.

**We agree with the reviewer that the records are not just next to the core site, however we consider southern/central Sweden to be within the Baltic Sea region, we added the location of these sites in Fig. 1.**

Line 235. Strictly speaking, temperatures can be low or high, not cold.

**We changed the wording of this sentence in the revised manuscript.**

Line 241. Moros et al. (2002) did not suggest that the Baltic Sea experienced "a large regression" at 10.2 ka – they only suggested a regression. The evidence for this regression was weak. It is currently debated if the Ancylus Lake stage ended with a large regression, a small regression or no regression. If there was a indeed a regression in the Arkona Basin, it was definitely not caused by "a continental uplift".

**We deleted this sentence in our revised manuscript.**

Line 247. The authors suggest that a meltwater pulse occurred at 10.2 ka. However, some studies indicate that the Scandinavian Ice Sheet expanded at ca. 10.2 ka (the Erdalen event).

**With our results, and those of Weiss et al. (2020), a large contribution of meltwater is a possible explanation for the negative *n*-alkane and alkenone hydrogen isotope values noted prior to 10.2 ka. The Erdalen event mainly concerned the western Norwegian glacier, which might explain why the Arkona Basin sediments did not record such an event. Previous studies suggest a decrease of the Scandinavian Ice Sheet at that time (Muschitiello et al., 2015; Cuzzone et al., 2016). The decreased amount of ice (and therefore ice melt) relative to precipitation as source**

**water for plants is reflected in the shift to higher hydrogen isotope values. We have emphasized this point in the revised manuscript.**

Lines 258-259. did you observe a thin layer rich in remains of terrestrial plants at the same level? Such a layer is seen in many cores from the Arkona Basin.

**We did not find a layer that was rich in terrestrial plant remains at the same depth.**

Line 266 Pinaceae is a family name, it should not be in italics

**This has been corrected in the revised manuscript.**

Line 267 "can be tentatively attributed to *Juniperus* shrub extension" (should be expansion?). To my knowledge, no pollen records from the region show a *Juniperus* peak at ca. 9.2 ka. However, the 9.2 event was short-lived and you need extremely high-resolution pollen analyses and high sedimentation rates or varves) to be able to detect possible influence on the vegetation.

**We revised the sentence to say "expansion" rather than "extension" (L363). We removed the discussion about the 9.2 ka event.**

Line 268. I don't understand why the authors chose to compare their record with pollen records from far away (northernmost Finland and Bohuslän in south-central Sweden). Why not compare with nearby pollen records? Anyway, to my knowledge no maximum occurrence of *Pinus* and *Juniperus* at 9.2 ka have been reported in pollen diagrams from the Arkona Basin region. See for example the detailed and well dated pollen diagram from Krageholmssjön in Scania in southernmost Sweden (Berglund et al. 2008, Veget Hist Archaeobot).

**We added the two Berglund et al. (2008) findings into our discussion of regional vegetation. These two pollen records show an increase in *Pinus* between 9.2 and 9.5 ka (Berglund et al. 2008a – doi: 10.1016/j.quaint.2007.09.018, and Berglund et al., 2008b – doi: 10.1007/s00334-007-0094-x).**

Line 269. what is a regional lake?

**By "regional lakes" we mean lakes from the Scandinavian region. We added the location of the pollen records used for comparison in Fig. 1.**

Line 289. Not sure what you mean by this: "The global transition from a glacial to an interglacial climate state across the Holocene, was punctuated by a few abrupt cold events". The cold events mentioned in the following happened long after the glacial-interglacial transition.

**We agree with the reviewer, the actual transition between glacial/interglacial period happened before the Holocene. The cold events mentioned here occurred during the Holocene. However, we have deleted this sentence in the revised manuscript.**

Line 293. It is unclear to me whether the authors see evidence of the 9.2 ka event in their data.

**At 9.2 ka we see a peak in $\delta^2H_{alkane}$, which may have been caused by the 9.2 ka event, but we deleted this paragraph from our revised manuscript.**

Line 297. Moros et al. did not give an age of 7.7 ka for the re-establishment of the connection between the Baltic Sea and the North Sea. From where did you get this age?

**Moros et al. (2002) report an age of 6.475 ± 50 $^{14}$C yr BP for the transgression. Warden et al. (2016) provide the age of 7.7 ka for the start of transgression which reached a maximum at around 7.2 ka, this reference has been corrected (L377).**

Line 298. Do you mean that the onset of the transgression lasted from 7.7 to 7.2 ka. Or do you mean that the transgression lasted from 7.7 to 7.2 ka? I believe that the marine transgression of the Arkona Basin began somewhere between 7 and 8 ka and continued for the rest of the Holocene, although the transgression rate slowed down after ca. 6 ka.

**Based on the discussion in Warden et al. (2016) and the data from our study, the start of the transgression occurred around 7.7 ka and reached a maximum at around 7.2 ka. We clarified this in the revised manuscript (L377).**

Line 300. regional warming began already in the earliest part of the Holocene, although interrupted by some short-lasting cold events.

**We changed this in the revised manuscript.**

Line 304. change *n*-alkanes were to *n*-alkane values were.

**The s is now added to values.**

Line 341. Again, to my knowledge, no "large fluctuations in the extent of gymnosperm cover" have been reported by pollen studies from the region.

**We changed this in the revised manuscript (see the revised section 4.2.2).**

Line 345. "lack of diversification of terrestrial vegetation noted for this period". Which period? The Ancylus Lake stage? The pollen records from the region show that many species arrived during this period.

**Between ~11 and 6 ka, woody species dominate pollen assemblages in the region and there is a gradual increase in species diversification. We have revised this statement in the revised manuscript (L347-349).**

Line 347. "regional warming which continued into the Late Holocene". Warming certainly did not continue into the Late Holocene.

**We have removed "which continued into the Late Holocene".**

Line 351. I don't think that we can ever "fully understand the complexity of paleoenvironments" as stated by the authors – not even if we use multiple proxies.

**Perhaps this is true, but multiple proxies can certainly provide a way of better understanding this complexity.**

Figures and tables

The authors have Age (ka), Age (Ka) and Age (kyr). I am not sure if the journal has a style to follow, but it should be consistent.

**We have made sure to consistently use the same notations throughout the revised manuscript.**

Table 2 and 3 can go to supplementary.

**We have added these tables to the SI.**

**Response to reviewer 2**

The article by Weiss et al. is well written as well as well illustrated and the record is based on a core from a particularly interesting region of the Baltic Sea for which new biogeochemical data is to be appreciated. I am not an expert on this kind of data myself, but think that this publication very well and precisely documents the used approaches, and I think that the results and the discussion are worth being published to broaden the view on the Holocene in the southern Baltic Sea, even though the relatively low temporal resolution of the record makes detailed comparison with other records difficult (see below). I also think that the age model could be explained in more detail since one has not only to visit the supplementary material of Weiss et al. 2020 but ideally also Warden et al. 2016 to get a precise idea how it was generated. Another aspect I see a little sceptical are the results concerning the terrestrial ecosystem structure since I think that with the used proxies, it can only be well estimated concerning a few certain plant types (such as *Sphagnum* indicated by the $C_{23}$ n-alkane). In case of several statements concerning the vegetation/climate development, I think the literature cited for comparison is only partly suited to support the results based on the biogeochemical proxies – probably better references for direct comparison can be found (see below).

**We thank the reviewer for these positive comments. As mentioned in response to reviewer one we included more detail on the age model (L130-150) as well as the vegetation changes in the region (Section 4.2.2) in the revised manuscript.**

**The dynamics of *Sphagnum* specifically can be difficult to determine in a mixed (sub)Arctic and temperate drainage basin as the dominant chain length of the *n*-alkanes has been reported to shift from $C_{23}$ in temperate environments to $C_{31}$ in Scandinavian (sub)Arctic environments (see Vonk and Gustavsson, 2009).**

Detailed remarks

Title and Abstract: I personally think that "structure" in the title implies more than the paper can provide in the end. Consider that in the abstract only a water source shift and a suggested vegetation diversification are mentioned concerning terrestrial ecosystem structure. General, the abstract could give less methodology and more own environment-related results.

**We have changed the title to better reflect the structure and content of the manuscript into: "Co-evolution of the terrestrial and aquatic ecosystem in the Holocene Baltic Sea". The abstract has also been adapted to focus on the results.**

 Sections 1 to 3.2

I have almost no remarks to sections 1 to 3.2, because they are generally very well written and are of appropriate length and focus. One remark only: In line 33 the expression "the SIS melted, exposed the land …" sounds a little odd to me.  I cannot say very much concerning the method sections since this is out of my expertise, but as far as I can tell this is also well done.

**We rephrased the sentence on (L42: "Melting and calving stimulated retreat of the SIS") in the revised manuscript.**

Section 3.3

I am not sure if there is an inconsistency with the discussion here – maybe it should be mentioned here already that the C28 1,14-diol was present in high amounts during the MB phase as mentioned in section 4.4 - the expression "only present" does not imply high amounts in my opinion.

**We have added that the 1,14-diols (including the C$_{28}$ 1,14-diol) were present in relatively high amount in the MB phase to section 3.3 of the revised manuscript, L253-254.**

Line 199: Checking Kotthoff et al. 2017, it appears to me that the Yoldia Sea (YS) is not reflected in the record described there, a comparison should not start before ca. 8000 yr BP.

**As mentioned in our response to reviewer one, we have included a brief discussion of Kothoff et al. (2017) because it is the only other study on long-chain diols in the Baltic Sea. We have made it clear in the revised manuscript that the comparisons of the two records are from 7.4 ka onwards.**

Section 4

Section 4.2 line 234 and following: it is implied here that the pollen records mentioned in line 236 are nearby, but I think the records used un Seppä and Birks 2001 are quite far away (some pollen records from northern Germany, Southern Sweden, Denmark or Poland would be nearer).

**We agree with the reviewer that the records are not adjacent to the core site, however we consider southern/central Sweden to be within the Baltic Sea region (See Fig. 1). In addition, the whole drainage basin of the Baltic Sea should be considered when assessing *n*-alkane data as they can be transported by rivers over long distances. We have added the two Berglund et al. (2008) findings into our discussion of regional vegetation (Section 4.2.2).**

Section 4.2.2

First paragraph: Again, I am not sure if Seppä and Birks 2001 is well fitting here.

**We have added several other pollen records to the discussion. The locations of these records can be seen in Fig. 1.**

In line 269 in the same paragraph it is stated that "the maximum extension of *Pinus* and *Juniperus* was recorded at 9.2 ka in these regional lakes"… Checking the cited literature, I can only partly agree: There are only minor increases of *Juniperus* pollen in the related time interval in Seppä et al. 2005/Digerfeldt 1977 and the *Pinus* peak is earlier. In Antonsson and Seppä 2007, there is a peak in *Pinus* percentages at 9.2 ka, but *Juniperus* percentages are significantly higher during the late Holocene and one could not speak of a maximum extension of this taxon around 9.2 ka. The sentence in line 270 and the following lines is correct concerning *Alnus*, but the pollen diagram of Lake Trehörningen depicted in Antonsson and Seppä 2007 does much more imply a decrease in *Pinus* (this one is very clear) than in *Juniperus* percentages (which does not seem to be consistently present between 11 and 5 ka anyway). I would think in this context that the attribution of the C isotope shifts to *Juniperus* shrub extension is not supported by the cited pollen records.

**We have noted that regional studies do not always find an increase in gymnosperms (L365-366). In the revised section 4.2.2, we do not discuss any maximum extensions, but rather an expansion of *Juniperus*. We hypothesize that *Juniperus* expansion is driving the change in C isotopes at this time because *Juniperus* produces higher amounts of *n*-alkanes than *Pinus*.**

It would be really nice to have closer pollen records for comparison in which coeval *Juniperus* increases were present to support the interpretation concerning the C isotope shifts (e.g. the record

shown by Yu et al. 2005 is quite near, but does not reveal such a signal), or if possible to see pollen data from core 64PE410-S7 itself.

**Unfortunately, we do not have a pollen record for this core, however, we added more sites to our comparison (Fig. 1).**

Line 294: A citation for the 8.2 ka event would be good, particularly if one could be found for the research area.

**The discussion of the 8.2 ka event has not been included in the revised manuscript.**

Conclusions:

Here, the 8.2 ka event is mentioned again (and marked in the figures, too) while it was said in 4.2.2 that no such event could be found in the own record and no citation was given in the discussions for such an event. If mentioned in the conclusions, this should be discussed in more detail and with a citation that there was indeed a cold event around that time in the research area. Generally, it would be good if conclusions based on own results would be better indicated.

**Following reviewers' comments, we have removed the discussion of the 8.2 ka event.**

**Response to reviewer 3**

This is a review for the manuscript "Co-evolution of terrestrial and aquatic ecosystem structure with hydrological change in the Holocene Baltic Sea" by Weiss et al. The authors use a large suite of organic biomarker proxies to analyze environmental change in the Arkona Basin for the Holocene. As expected, large changes in the different proxies indicate large changes in hydrology and possibly climate in the region fitting with the well-known different phases of marine conditions in the Baltic. However, I do have some problems with the current structure and missing discussion. The resolution of the records is very low, in several cases there is only two or three datapoints for a phase. Along with the absence of any information on the age model, this makes a discussion on trends in phases and at their transition and the presence or absence of events like 8.2 or 9.2 kind of useless. The discussion basically reads as a long list, i.e. "this proxy changed in this direction meaning that" without hardly including anything on the many studies in the area itself; I think the ms is missing a big opportunity to make this a much better story (see also the comments below).

I am not an organic biomarker specialist so I cannot judge on the suitability of the methods, although the description of the analyses and their background in the introduction is very elaborately done.

**We thank the reviewer for their constructive comments. The two other reviewers also asked for more information about the age model, and we added a new paragraph in the revised version (L130-150).**

I'm missing a discussion that is involving the enormous amount of studies that have been performed in the Baltic already. Most references are only related to biomarker records, some of them from non-Baltic locations. Because the resolution and age control are low and you are focusing on different phases, my suggestion would be to restructure the ms by starting to identify your different phases and what they are based on (i.e. existing literature) and then pool the samples you have for those phases into a specific signal for that phase so that you are basically creating snapshots of those phases. In a next step these can then be compared with studies that are especially concentrated around the southern Baltic like the Arkona/Bornholm area (IODP expedition 347 – Site M0065; Heinrich et al., 2018; Anjar et al., 2012; Jensen et al., 2016); Belt seas and Kattegat (e.g. Kotthoff et al., 2017, Ni et al., 2020; Hyttinnen et al., 2020), or lake/terrestrial records from northern Germany and southern Sweden

(e.g. Dräger et al., 2017; Hannon et al., 2018). If such studies can be linked with the biomarker results it would make the study much more valuable in identifying the processes behind the signals.

**Thank you for this suggestion. We have restructured the discussion and focused on the phase-specific changes. We have added more information about vegetation from lacustrine pollen records to help create the snapshots of each phase as recommended.**

Line 31: The Baltic Basin existed long before the deglaciation

**We have acknowledged this on L32-33: "The Baltic Sea existed at least as far back as the Eemian interglacial (~130 ka, Andrén et al., 2015)."**

Intro: A very detailed background on organic proxies, but nothing about other salinity proxies in the Baltic. Many studies have attempted to reconstruct salinity changes in the Baltic, e.g. Gustafsson and Westman, 2002; Emeis et al., 2003; Mertens et al., 2012; Ning et al., 2017, Ni et al., 2020 and others.

**We have added some discussion on this topic. The revised manuscript now reads: "The complex climate dynamics caused substantial shifts in the salinity of the Baltic Sea during the Holocene, as indicated by changes in the phytoplankton population (e.g., Alhonen, 1972; Weiss et al., 2020). The timing of different Holocene phases of the Baltic Sea is debated and appears to be divergent at different locations in the basin (Bjorck; Gustafsson and Westman, 2002; Moros et al., 2002)."**

First paragraph of the intro could use more referencing, it's very well studied!

**More references have been added to the introduction.**

Terrestrial vs marine….how does that work in the Baltic? How do you define the brackish environments with this regard?

**The comparison should be between terrestrial and aquatic rather than marine in the case of the Baltic. The basin varied between freshwater and marine conditions. This has been changed in the revised manuscript.**

Age model: Just a short reference to previous papers explaining the age model is not enough. Sedimentation rates and variations can be extreme in the Baltic especially when changing between the different settings. It is essential that this is explained and shown in the ms. The Arkona Basin is located at an interesting point just after where the saline inflows enter the present Baltic. Timing in this area does not necessarily have to be the same as in the Baltic Proper or the Straits/Kattegat. Simply assuming that this is the case is unlikely to be true. Same goes for comparison with the lake records in Finland.

**We have added a new paragraph explaining how the age model was constructed (L130-150).**

I would not call the first 9.30 meters the top of the core when the whole core is 12 m. A 100 cm resolution is very low; is there any particular reason why this is so slow when downcore records were going to be reconstructed?

**Our focus was primarily on the Ancylus Lake phase and the transition to more marine conditions, hence the higher resolution sampling in that part of core.**

Sediments in the Baltic are notorious for transport of material. How does this affect the different proxies? Radiocarbon studies in the central Baltic have shown that organic matter is continuously re-deposited and can result in large temporal differences. Could this the reason some of your changes are not aligning with commonly accepted events?

**We cannot entirely rule out redeposition as an explanation for the smoothing out of some major events. However, the proxies used here should be affected by deposition in a similar manner to micropaleontological and other proxies used to establish such events in the Baltic. Those proxies record commonly accepted events, thus we have reason to believe the proxies used in our study should as well. The alkenone distributions and changes therein line up very well with known Baltic Sea phases. The large positive shift in alkenone hydrogen isotope values takes place in the middle of the Ancylus Lake phase where alkenone distributions were stable (Weiss et al., 2020). The positive shift in n-alkane hydrogen isotope values takes place at the same time suggesting it is not linked to a shift in Baltic Sea phase, but something that changes both the hydrogen isotopic composition of water in the terrestrial and aquatic realms simultaneously. A change in relative contributions of isotopically light meltwater versus more $^2$H-enriched (heavier) precipitation in the form of rain is the most likely explanation for such a large shift. This is confirmed by regional pollen records (which suggest an increase in precipitation at this time) and reconstructions of the SIS (which suggest that the ice sheet was great diminished).**

Line 241: With continental uplift I assume you mean isostatic rebound?

**Yes, we do mean isostatic rebound.**

Lines 246-248: "The SIS was retreating at this time (Muschitiello et al., 2015; Cuzzone et al., 2016), thus it is plausible that a meltwater pulse transported a higher concentration of n-alkanes from the north into the basin just after 10.2 ka." This reads like kind of a loose statement. This would require things like age control, and are there signs of a meltwater pulse (e.g. sedimentological)? Has this been shown before, then cite it, and if not you need to bring more explanation.

**The indication for the relatively high meltwater contribution comes from the relatively low hydrogen isotope values. The increase in *n*-alkane hydrogen isotope values at 10.2 ka suggests a reduced contribution from $^2$H-depleted (lower hydrogen isotope composition) meltwater to the system.**

Section 4.2.1: First you conclude that the low concentration of alkanes indicates less continental runoff, but then following the other proxies you conclude that more continental runoff occurred. What does the literature say about this? Which pathways, e.g. rivers, were in the area, maybe climate was actually changing becoming drier or wetter.

**In the early part of the Ancylus Lake, which we refer to as the "ice-melt subphase", the *n*-alkane concentrations are low (likely due to ice cover). Towards the end of the ice-melt subphase, there is a large increase in *n*-alkane concentrations. We link this to an increased contribution of meltwater that signals the end of the SIS influence on regional hydrology.**

Lines 293-295: "While our record is of insufficient resolution to capture this rapid event, the increase of δ2H alkane values noted at 9.2 ka is presumably also influenced by the environmental conditions present at that time. Another rapid cold event occurred at 8.2 ka, which is not observed in our record, but may be elucidated with higher resolution sampling at this time interval." Indeed, as the age control is lacking and the resolution low your 9.2 event may well be the 8.2 one.

**As mentioned in response to reviewer 1, we note a peak in d2Halkane at 9.2 ka which may have been caused by the 9.2 ka event. Since we cannot be 100% sure, we have deleted this paragraph from the revised manuscript.**

Several of the curves in the figures have no error bars on them. It would be good to add them. Add other relevant study sites to the map and include them into the discussion, e.g. Bornholm, northern Germany, southern Swedish lakes, Little Belt.

**The error bars seem to be missing because they are small in some cases. The errors are listed in the tables (which will be moved to the SI in the revised version). Locations of pollen records discussed have been added to Figure 1.**

The Pangaea link is still missing.

**It will be added once the data is available on Pangaea.**

To conclude, I think this dataset definitely has the potential to make an interesting manuscript after restructuring and expanding the discussion. So currently I recommend major revisions.

**We appreciate the constructive feedback and have incorporated it into the revised version of the manuscript.**

---

## Author Response (AR2)

Dear Dr. Seidenkrantz,

Thank you for the opportunity to revise our manuscript, "Co-evolution of terrestrial and aquatic ecosystem structure with hydrological change in the Holocene Baltic Sea" for publication in Climate of the Past. We appreciate your feedback and understanding throughout the review process and the constructive comments from three reviewers which have significantly improved the manuscript. To address the most significant issue, we have now included a figure of the age model. We have also addressed the remaining comments from the reviewers. Our response to reviewer comments can be found below. As requested, we are submitting both a clean copy of the revised manuscript and a separate document with highlighted changes.

We believe this improved version better presents the biomarker-based results in context with what is known about climate in the Baltic Sea region throughout the Holocene. We hope you will now be able to accept this revision for publication in Climate of the Past.

On behalf of all co-authors,
Gabriella M. Weiss

Dear Dr. Weiss,

Thank you for your re-submitted version of your manuscript "Co-evolution of terrestrial and aquatic ecosystem structure with hydrological change in the Holocene Baltic Sea" to "Climate of the Past". As you are aware, your manuscript has been evaluated by three reviewers; all where the same as for the first review. All reviewers feel that you have significantly improved your manuscript and have to a large extend followed their comments.

However, as also pointed out by some of the reviewers, one important issue still remains: the age model. You now better explain, how you constructed the age model; however, as you do not show any figures on this, the reader has no chance of evaluating your age model. Thus, it is imperative that you present the age model.
You explain that you constructed the age model via a combination of 14C dates on your own core and correlation of the XRF data with two existing cores. However, you neither provide the 14C dates, nor show the actual data used in the correlation. It is imperative that this information is added. I strongly urge you to 1) add a table with the dates in the standard format (lab no, core depth, material, dates, errors, calibration, d13C if available) and 2) add a figure where you show the actual correlation of the data. You could make a figure where you show a picture/lithological log as well as the Ca/Ti and Br data from your own core (including marking the levels of your 14C dates) to the left and the data as well as dates from the two existing cores to the right. The correlation (wiggle matching) of course needs to be shown through correlation lines.
You must also 3) show the actual age model created through the Bacon software based on the 14C dates and XRF tie points. It is possible, that the age model can be combined in the same figure as the correlation figure above.
If you end up with an overload of figures, you could easily combine Figs 4-6 into one figure with 4 sub-figures.

**We have included a figure of the age model – XRF data with tie points, 14C dates, and the age model generated by Bacon.**

In addition, Reviewers 1 and 2 indicate a few further, but minor corrections and comments that you should take into accounts.

As always, when you resubmit, please reply to all reviewers' comments in detail and mark clearly any changes made to the manuscript in the text through highlights or track-changes.

I look forward to seeing your revised manuscript.

Kind regards,
Marit-Solveig Seidenkrantz
co-Editor in Chief, Climate of the Past

Non-public comments to the Author:
I have marked the decision as "minor corrections", as, if I am satisfied with your improvements, I may not send the manuscript out to review again. However, please note that I will not accept the manuscript for publication unless you present your age model, so in this respect, it is in fact a

major, imperative correction. I know that making new figures is time consuming, but it will also significantly improve your paper.

**We appreciate your understanding and have now included a figure with the age model.**

**Reviewer #1**
Comments to manuscript: Co-evolution of terrestrial and aquatic ecosystem structure with hydrological change in the Holocene Baltic Sea

I have commented on an earlier version of this manuscript and suggested major revision. I can see that the authors have followed most of my suggestions as well as the suggestions by the other referees.

**Thank you for the constructive suggestions that helped to improve the manuscript.**

One point I raised was the question about the age-depth model. The authors have added some notes on this issue, but they still do not present an age-depth model. It is still unclear to me how the older non-marine part of the core was dated. The authors note in the manuscript that the "age model was created by combining 14C-ages of mollusk shells and correlation of Ca/Ti and Br records with two nearby cores… (Warden et al., 2016). However, Warden et al. did not date the non-marine part of their cores. Instead, they referred to Moros et al. (2002). Moros et al. referred to Björck (pers. comm.). The most honest would be to plot data against depth, but this makes it difficult to compare and discuss with other records. Anyway, I think that the need to mention that the chronology of the older part of their record is uncertain.

**We have now included an age model figure (Figure 2) which shows the XRF tie points and 14C ages.**

Other comments

Line 14. Ice melt, should be ice retreat

**We have changed "melt" to "retreat."**

Line 19. Western should south-western

**We have changed this.**

Line 19. "In the earliest part of the record (10-8.2 ka)". According to line 16 the record spans tha last 11 ka. Does the record span 10 or 11 ka?

**The record does go back to 11 ka, however 10 – 8.2 ka is the period where the hydrogen isotope values show a big change. We have revised this statement to say, "In the earlier part of the record (specifically 10 – 8.2 ka) …"**

Line 34. The Baltic Ice Lake started to form around 13 ka (Björck 1995). Please note that this reference refers to C-14 years BP. Therefore, it should be c. 15.5 ka, not 13 ka.

**Thank you for catching this. We have changed the date to 15.5 ka.**

Line 34. The Scandinavian Ice Sheet retreated (by melting and calving). Retreated is more correct than melted.

**The addition of "melting and calving" was suggested by another reviewer. We choose to keep the parenthetical note in.**

Line 49. Continental uplift, it should be glacio-isostatic rebound.

**We have changed this phrasing as suggested.**

Line 57. Deciduous woody forests, suggest change to deciduous forests.

**We have removed the word "woody."**

Line 90. There are no C4 or CAM plants in the region, therefore the discussion about such plants are not relevant.

**We have removed this statement.**

**Reviewer #2**
*No comments, suggested to accept as is.*

**Thank you!**

**Reviewer #3**
Weiss et al. have extensively responded to the comments from the three reviewers and significantly improved the manuscript. The division of the discussion into the different phases improves the clarity a lot. What I would still suggest though is to add the ages you use for these phases, either in a table (where you could also add the previously determined ages for these phases and/or per area, e.g. Baltic Proper, Arkona/Southern Baltic, Belt Seas/Kattegat, or simply between brackets with each header. The reason being that, as you also point out, these phases may have been slightly different in different areas of the Baltic. I would also consider to change the title further to include the Ancylus Lake already with ages in brackets, because that is the main message of the paper. The rest of the Holocene has a few data points too, but is not adding that much to the story.

**Thank you for your comments that have helped improve the structure and flow of the manuscript. We have now added ages in brackets to the headers in the discussion section.**

I think the introduction can still be improved. From my first review I gathered that background on salinity was missing as the main reconstruction using the organic proxies was to reconstruct

salinity. But now an extensive paragraph is added on the vegetation history. This is definitely a good overview of what is available in the area but it also makes me wonder a bit what the organic proxies are adding to this. You mention the organic proxies are to complement the existing pollen records. So is this then adding information because they are more sensitive to hydrological changes than the pollen records, as these are not that much variable in these areas? I think you can bring this in the introduction more clearly as a statement that using the organic proxies you can separate between, e.g., meltwater and precipitation phases.

**We have added a sentence to lines 77-78 that states, "Organic compounds and their isotopic signatures can provide additional constraints on changes in hydrology not easily discernible from pollen assemblages (e.g., they allow for differentiation between ice melt and precipitation as a water source)."**

Presentation of the age model is still a major issue. All three reviewers pointed this out and the authors responded in detail. But I can still not find where these data are presented. I suggest adding this correlation with the other sites in the supporting information. Working with these different phases in the Baltic makes the age model essential, and when the reader then has to go find another paper or even the supporting information to those other papers, is not very convenient. The Warden paper does not have bromine data in it, and in the Weiss 2020 paper I can only find an excel table as supporting information that has the XRF data in it, but nothing on the age model. It would also be good to provide an error estimate on the ages, i.e. using the 9.2 ka as boundary between the two sub-phases of the AL needs an uncertainty range on it.

**We now include a figure with the age model (Figure 2) which contains XRF tie points, 14C dates, and the age model output by the Bacon software.**

Lines 204-206: "The large variations in sedimentation rates for core 64PE410-S7, and the nearby cores to which it has been correlated, are likely related to the shallow water depth in the Arkona Basin". Why is that?

**We have revised this sentence to emphasize that the fluctuating connection with the North Sea caused the large changes in sedimentation rates in the shallow Arkona Basin. "The large variations in sedimentation rates for core 64PE410-S7, and the nearby cores to which it has been correlated, are likely related to changes in the connection between the North Sea and the otherwise shallow Arkona Basin (~45m)." (Lines 213-215)**

Lines 406-410: Why would this indicate that these temperatures would be unreliable? I think this may simply point to a signal that is mainly produced during a specific season, e.g. spring.

**We have added this information (now on line 419), "This indicated that reconstructing temperature using LDI in the MB may be unreliable, likely due to the influence of diatoms and freshwater input, or that LDI may be reflecting a specific season (e.g., spring)."**

In summary, I think the manuscript has improved a lot but still needs some streamlining and the information on the age model definitely needs to be included. After that, the manuscript would be a valuable addition to Climate of the Past.

**Thank you for your constructive comments that have helped improve our manuscript.**